# EIBench: Assessing the Emotion Interpretation ability of Vision Large Language Models

## Abstract

Affect computing is crucial in fields such as human-computer interaction, healthcare, and market research, yet emotion's ambiguity and subjectivity challenge current recognition techniques. We propose **Emotion Interpretation** (EI), a task that interprets the reasons behind emotions, and create the **E**motion **I**nterpretation **Bench**mark (EIBench) using a VLLM-assisted dataset construction method, Coarse-to-Fine Self-Ask (CFSA), with carefully human in-the-loop annotation. EIBench includes 1,615 basic and 50 multi-faceted complex emotion interpretation samples. Experiments show limited proficiency of existing models in EI, with the best achieving 62.41% accuracy in the zero-shot setting and some performing lower than the text-only LLaMA-3 model (6.26%) in the caption-provided setting. Different personas assigned also differ the benchmark results. Overcoming the challenges posed by EI can result in more empathetic AI systems, thereby enhancing human-computer interaction and emotion-sensitive applications.

## 1 Introduction

Affect computing plays a crucial role across diverse domains (Khare et al., 2024), such as human-computer interaction (HCI) (Jain et al., 2023; Ma et al., 2022; Parviainen & Søndergaard, 2020; Yang et al., 2019), healthcare (Dahl & Harvey, 2007; Saarni et al., 2007; Tronick, 2018), and market research (Cambria et al., 2017; Caruelle et al., 2022; Srivastava & Bag, 2024). Recent research mainly focuses on recognizing emotions by categorizing them into basic types. However, the inherent complexity and subjectivity of emotions make it difficult for individuals to accurately identify their own emotions in complex situations. Despite this, these applications share a common goal: understanding the triggers of emotions. The variability of emotional experiences across individuals and contexts underscores the need to move beyond simple categorization and focus on the triggers and circumstances that lead to emotions, known as "Emotional Triggers".

In response to this motivation, we propose the ***Emotion Interpretation*** (EI) task, which focuses on interpreting emotional triggers rather than categorizing emotions. As shown in Figure 1, this task involves identifying the specific causes of emotional states for given individuals or scenes, aiming to better understand human's emotion by try to think at their own position. Overcoming the EI task can result in a more empathetic AI system and applications. The Vision Large Language Models (VLLMs), known for their extensive world knowledge and explanatory abilities, are well-suited for emotion interpretation tasks (Bai et al., 2023; Chen et al., 2023a; Lin et al., 2023; Liu et al., 2024b; Wang et al., 2023; Liu et al., 2024a; 2023; Li et al., 2023b) But due to the lack of existing benchmark, the performance still can not be measured.

To advance research in this area, we established the **E**motion **I**nterpretation **Bench**mark (EIBench) (Figure 1), which includes 4 primary emotions of 1615 basic EI samples, and 50 complex multifaceted EI samples, such as combinations of happiness and sadness (Figure 1 (e)). We also proposed an VLLM-assisted data annotation scheme, the Coarse-to-Fine Self-Ask (CFSA) method, which employs the Chain of Thought (CoT) approach (Press et al., 2022; Madaan et al., 2024; Yao et al., 2024; Besta et al., 2024; Zhang et al., 2023; 2022) to guide LLMs in preliminary annotation.

We conducted a comprehensive assessment of commonly used open-source and closed-source models to thoroughly evaluate different models' abilities in the EI task. There are 4 evaluating settings: 1) zero-shot to user questions, 2) combining image captions with user questions, and 3) reasoning with CoT, 4) LLM under different personas. The experiments indicate that existing models

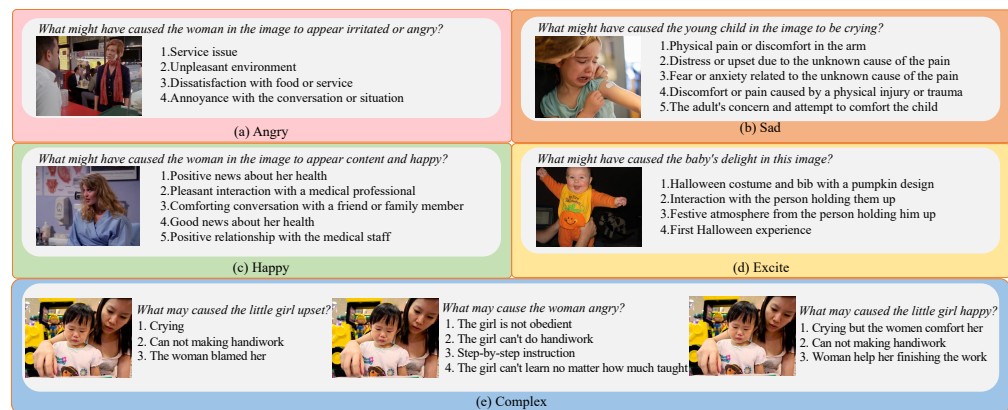

Figure 1: Figure (a-e) represent the different emotion categories under five scheme: angry, sad, excite, happy, and complex respectively, and the Emotion-trigger pairs.

still have significant shortcomings in interpreting emotions, with closed-source models generally outperforming open-source models. Interestingly, the evaluation results for basic and multifaceted emotions deviate from the expected pattern. The Claude-3 series, which performed best on the basic subset, underperforms the ChatGPT-4 series on the multifaceted complex subset. This phenomenon might be due to the Claude series' tendency to provide definitive answers, whereas ChatGPT-4 demonstrates better capability in handling multifaceted reasoning.

Our contributions include: (1) proposing the Emotion Interpretation task, which focuses on interpreting emotional triggers rather than merely classifying emotions, crucial for building more empathetic systems; (2) introducing the Coarse-to-Fine Self-Ask (CFSA) method, an effective VLLM-assisted data annotation technique; (3) developing the EIBench dataset, which includes 1,615 basic EI samples, and 50 multifacets complex EI samples; and (4) extensively evaluating both open-source and closed-source models, highlighting the limitations of current models in interpreting emotions.

## 2 RELATED WORK

We review the relevant literature in emotion recognition, emotion cause extraction, humor studies, and chain-of-thought prompting techniques, providing context for our work on interpreting emotion.

### 2.1 CONTEXT AWARE EMOTION RECOGNITION

Context Aware Emotion Recognition (CAER) goes beyond Facial Expression Recognition (FER), which focuses solely on perceiving emotion via the face (Wang et al., 2020b;a; Vo et al., 2020; Zheng et al., 2023; Mao et al., 2023; Li et al., 2023c; Cheng et al., 2023). CAER considers the emotional cues provided by background contexts, integrating facial and body language in a joint and boosting manner (Kosti et al., 2017; Yang et al., 2023a; Xenos et al., 2024; Bhattacharya et al., 2020; Ruan et al., 2020; Mittal et al., 2020; Li et al., 2021; Yang et al., 2022; Zhang et al., 2019). Various methods have contributed to this area. For example, (Kosti et al., 2017) established the EMOTIC dataset and proposed a baseline that combines the body region and the whole image as the context. (Lee et al., 2019) proposed a dataset derived from movies with human social context. (Yang et al., 2023a) built a context dictionary based on clusters of visual features to identify meaningful patterns using visual context. Additionally, (Xenos et al., 2024) explored CAER with the commonsense knowledge from VLLMs, achieving SOTA on EMOTIC (Kosti et al., 2017) and CAER-S (Lee et al., 2019) datasets.

### 2.2 EXPLAINABLE EMOTION RECOGNITION WITH LLMS

Large Language Models (LLMs) have been increasingly used for emotion recognition due to their world knowledge (Ouyang et al., 2022; Liu et al., 2021; Fei et al., 2023; Lei et al., 2023). In NLP study, (Fei et al., 2023) used Chain of Thought (CoT) prompting to recognize implicit emotions, while (Lei et al., 2023) designed a retrieval-based system for conversational emotion recognition. The development of Vision LLMs (VLLMs) (Liu et al., 2024b; Lin et al., 2023; Liu et al., 2024a) has further expanded the application of LLMs in emotion recognition (Cheng et al., 2024b). However,

due to the lack of emotion-related datasets, the capability of LLMs in this area is limited. To address this, (Xie et al., 2024) employed visual instruction tuning at various emotion data for better performance, (Xenos et al., 2024) used VLLMs to extract commonsense context, combining it with image data to train a transformer model, and (Fei et al., 2023) utilized the CoT approach to incrementally guide models through emotion tasks. While these works focus on identifying emotion types, this research aims to interpret the implicit emotion triggers behind human emotions, comprehending the formation of emotions.

## 2.3 HUMOR STUDY

Humor is an integral part of human life and has been the focus of extensive research (Chandrasekaran et al., 2016; Hwang & Shwartz, 2023; Hessel et al., 2023; Hyun et al., 2023; Chen et al., 2023b; Hasan et al., 2019; Yang et al., 2015; Annamoradnejad & Zoghi, 2020; Hasan et al., 2021). For instance, (Chandrasekaran et al., 2016) examined the elements or characteristics within cartoon scenes that contribute to humor. Similarly, Memecap (Hwang & Shwartz, 2023) compiled a dataset of 6.3K visual memes with visual metaphors to facilitate meme interpretation. (Hessel et al., 2023) tested the ability of large language models (LLMs) to understand humor using a subset of the New Yorker Cartoon Caption Contest. Additionally, (Hyun et al., 2023) introduced the Video Laugh Reasoning task to explain why people laugh in specific videos. (Chen et al., 2023b) investigated pretrained LLMs' ability to understand Chinese humor, which may also aid future research in humor generation. Humor study delves into the elements that evoke laughter or amusement, interpreting the specific triggers of humor. Our EI task aims to interpreting the triggers behind a broader spectrum of emotional responses, not limited to amusement but extending to various emotional states.

## 2.4 EMOTION CAUSE EXTRACTION

Emotion Cause Extraction aims to identify the triggers lead to emotions was first proposed in text domain (Lee et al., 2010). For a more accurate identification, researchers propose Emotion Cause Pair Extraction (ECPE) that use multi-task learning frameworks to simultaneously predict emotions and their causes (Xia & Ding, 2019). Wang et al. (2024) further push it into multi-modal domain at SemEval challenge in conversation, for using multimodal input to identify the corresponding emotion trigger with current speaker. Zhang et al. (2024); Cheng et al. (2024b) won the challenge by using powerful LLM based method InstructERC (Lei et al., 2023) for the context understanding. Our task builds upon ECE for we not only looking at what can be found in the input explicitly, but more behind the picture. The EI task not only identify the triggers but also interpret them, needing deeper understand of the whole context, as well as the common sense knowledge.

## 2.5 CHAIN OF THOUGHT PROMPTING

Chain of Thought (CoT) Prompting enhances problem-solving by breaking complex tasks into manageable, sequential steps, improving both accuracy and transparency (Press et al., 2022; Madaan et al., 2024; Yao et al., 2024; Besta et al., 2024; Zhang et al., 2023; 2022). (Press et al., 2022) proposed the Self-Ask method, where LLMs generate and answer their own sub-questions to solve a larger problem. (Zhang et al., 2023) extended CoT to multimodal tasks, using a two-step approach where LLMs first generate a rationale and then use it for reasoning. (Zhang et al., 2022) introduced a method for LLMs to solve tasks step-by-step in a one-by-one manner. We propose a Coarse-to-Fine Self-Ask method to guide VLLMs in assisting the EIBench annotation. This method progresses from general to scenario-specific perspectives, gradually deepening the understanding of emotional triggers, thus extending the application of CoT techniques to the domain of emotion interpretation.

## 3 PROBLEM DEFINITION

**Task Definition** We list the emotion related task in Table 1, for the emotion recognition based task, the main goal is to predict the emotion $E_{emotion}$. Facial Expression Recognition (FER), the face information $X_{face}$ is the only input. Context Aware Emotion Recognition (CAER) considers the context $C_{context}$ together with $E_{emotion}$ in the prediction making. Emotion Recognition with LLM, especially those using CoT techniques or doing reasoning, dose not simply output $E_{emotion}$ but provide a series of intermediate output $Z_{mediate}^{1...n}$ before the final prediction $E_{emotion}$.

The Humor Study (HS) is likely a subset of the EI, for it aiming at understand or interpret the triggers of humor (Hessel et al., 2023) given a humor figure or text $H_{humor}$ to figure out the interpretation

Table 1: Comparison of Emotion-related Tasks. FER, CAER, ECE, HS, and EI stand for Facial Emotion Recognition, Context Aware Emotion Recognition, Emotion Cause Recognition, Humor Study, and Emotion Interpretation.

| Task | Descriptions |
|------|--------------|
| FER | Identifying emotion types based solely on facial information ($X_{face} \rightarrow E_{emotion}$). |
| CAER | Identifying emotion types based on facial and context information ($[X_{face}, C_{context}] \rightarrow E_{emotion}$). |
| ER with LLMs | Identifying emotion types based on reasoning ($[X_{face}, C_{context}] \rightarrow Z_{mediate}^{1...n} \rightarrow E_{emotion}$). |
| HS | Understanding the triggers of humor ($H_{humor} \rightarrow I_{humor}$). |
| ECE | Find the triggers of general emotions ($[E_{emotion}, C_{context}, X_{face}] \rightarrow T_{triggers}$) |
| EI | Interpreting the triggers of general emotions ($[E_{emotion}, C_{context}, X_{face}] \rightarrow I_{general\_trigger}$). |

$I_{humor}$ of it. For a pre-given $E_{emotion}$, Emotion Cause Extraction (ECE) task aims at find the triggers of general emotions in the given context only. Our task builds upon ECE for we not only looking at what can be found in the image, but for a boarder range for example what did not shows in the image (e.g. Figure 1 (e) whether the women blame at the child or comfort her) and a deeper understanding (e.g. Figure 1 (d) the text in the clothes indicating the first halloween experience).

To formalize, Emotion Interpretation focuses on understanding the underlying emotional triggers rather than identifying the emotion label. Given a query $q = (x, e)$ consisting of the input image $x$ (consist of face information $x_{face}$ and context $x_{context}$) and the emotion state $e$, a generative model $g$ can generate a trigger set $\mathcal{T}$:

$$\mathcal{T} = g(q) = g((x, e))$$

Here, $\mathcal{T}$ can be either a set of sentences:

$$\mathcal{T} = \{\text{"The person is sad because he lost his job.", "He received his notice of dismissal.",} \dots\}$$

or a set of labels:

$$\mathcal{T} \in \{\text{"job loss", "relationship issues",} \dots\}$$

The emotion state $e$ in the query $q$ can vary, being either positive (e.g., $e^+ = $ happy) or negative (e.g., $e^- = $ unhappy) for the same individual.

This task shares similarities with "Explainable Multimodal Emotion Reasoning" (EMER) (Lian et al., 2023), as both aim to provide explanations. EMER supports multi-class classification, but the nature of classification tasks limits its ability to output contradictory emotions.

Table 2: Emotional Trigger Types

| Atmosphere | Social Interactions | Body Movements | Facial Expressions | Objects |
|------------|---------------------|----------------|--------------------|---------|
| Performances | Outdoor Activities | Clothing | Sports | Other |

**Emotional Trigger** We define an emotional trigger to stimulus that provokes an emotional response in an individual (Table 2). These triggers can include atmospheres, such as a dimly lit room, and social interactions like arguments. Body movements and facial expressions, although not triggers themselves, play a crucial role in communicating emotions and can amplify the effect of other triggers. Objects with sentimental value, performances in music or theater, outdoor activities like hiking, clothing choices, and sports events all play significant roles in triggering emotions.

While many of these triggers can be visually depicted, there are also implicit triggers that can evoke emotions beyond the image, such as an athlete's adrenaline rush or the special feelings associated with people and scenes. We categorize these as "other" triggers. These varied triggers can deeply influence an individual's emotional state, often subconsciously, shaping their reactions and interactions.

## 4 EIBENCH

We refine the CAER-S (Lee et al., 2019) and EmoSet (Yang et al., 2023b) context-aware emotion recognition datasets using the CFSA method, assisted with human annotation, to create a benchmark dataset. To the best of our knowledge, this is the first benchmark focused on interpreting human emotions, featuring 1,615 basic EI and 50 complex EI samples.

Table 3: This table illustrates how our CFSA Method comprehends a person's excitement and joy during an LGBT event. Pink represents the generated caption, while yellow denotes the user query content, with both integrated. The light orange and trigger in the query are matched accordingly.

| An Example of Comprehending Excitement | | |
|---|---|---|
| | |  |
| User Question (VLLM) | | *What might have caused **the man holding the rainbow flag** to **become so excited and joyful**?* |
| User Question Preprocessing (VLLM) | Question Parse | Who is the person user talks about? The man holding the rain bow flag. What is the user's demand? To know why the man is so excited and joyful. |
| | Question Reconstruction | The man holding the rainbow flag is a proud and enthusiastic participant in a parade, possibly celebrating LGBTQ+ pride or a similar event. He is waving the flag high in the air, showing his support and joy. The user wants to know why the man is so excited and joyful. |
| General Self-Ask Thinking (Question: Text-only LLM (GPT-4)) (Answer: VLLM) | | 1. How many people in this image, what are they doing? ... They are standing on top of a box ... are likely participating in a parade or public event. 2. What does "the person user mentioned" do in this image? Does he have interaction with others? ... is holding the rainbow flag and waving it high in the air... 3. What is the facial expression of the person mentioned? The facial expression of the person holding the flag is one of excitement and joy. 4. How about his body language? ... holding the flag is expressive and energetic, ... active participation in the event. |
| Scenario Specific Self-Ask Thinking (VLLM) | | 1. What specific ... the man holding the rainbow flag ... relates to his excitement and joy? The man is celebrating LGBTQ+ pride ... as indicated by the rainbow flag. ... participating in such an event is a way for him to express his support for the LGBTQ+ community and celebrate their achievements and progress. 2. What factors, such as personal experiences or beliefs, might contribute to the man's enthusiastic participation in the parade or event? ... he may have friends or family members who identify as LGBTQ+ ... 3. How does the man's enthusiasm and joy impact the overall atmosphere and energy of the crowd and the event? The man's enthusiasm ... energy of the crowd and the event... 4. Are there any interactions or reactions from the crowd that might be influencing or being influenced by the man's excitement and joy? ... provide him with a sense of validation and encouragement for his participation in the event. |
| Emotion Summarization (Emotional Triggers) | | 1. Celebrating LGBTQ+ pride or a similar event. 2. Supporting the LGBTQ+ community. 3. Expressing his personal beliefs and values. 4. Feeling a sense of unity and belonging with the crowd. 5. Being part of a positive and uplifting event. 6. Standing on top of a box. |

## 4.1 VLLM-ASSISTED DATASET CONSTRUCTION

### 4.1.1 COARSE-TO-FINE SELF-ASK ANNOTATION

We develop a Coarse-to-Fine Self-Ask (CFSA) method (Appendix Figure 3) to assist the EIBench annotatation. CFSA involves breaking down complex, implicit user questions into a series of simple VQA queries. Specifically, the VLLM assistance annotation process can be decoupled into four phases: 1) initial question preprocessing, 2) general self-ask thinking, 3) scenario self-ask thinking, and 4) emotion summarization. Finally four volunteers conducted a thorough manual review and detailed annotation of the entire dataset at all the phases of annotation.

**Initial Question Preprocessing.** To comprehensively and automatically capture the visual context of the image related to the emotion, we utilize a fixed and simple prompt to stimulate the LLM to complete and rich the prompt for visual questioning. Technically, we first parse the initial prompt as: $s^{par} = \phi(s^{init})$, where $\phi$, $s^{init}$, and $s^{par}$ represent the GPT-4, initial prompt, and the parsed question (prompt), respectively. $s^{init}$ is constructed simply by a given emotion state $e$ and prompt template. After the rich prompt $s_{par}$ is produced, we collect the visual details rich reconstruction question as: $s_i^{rec} = llava(x_i, s^{par})$ $x_i \in \mathcal{X}$, where $llava$ and $x_i$ are the LLaVA-v1.6-34B (LLaVA-NEXT) (Liu et al., 2024a) VLLM and the input image, respectively. $s_{rec}^i$ denotes the reconstructed question produced by the LLaVA with given image $x_i$. The whole image dataset is denoted by $\mathcal{X}$.

Though VLLMs can describe images in detail, they may overlook emotional triggers due to limited emotion knowledge (Cheng et al., 2024a). However, with the right question prompts, their strong VQA capabilities can help uncover these triggers. We introduce general self-ask and scenario self-ask methods to guide this process.

**General Self-Ask Thinking.** We let GPT-4 generate open-ended questions for the entire dataset. Afterward, we identified the four most frequently asked questions to prompt the VLLMs. Specifically, we compile these questions into a set $\mathcal{S}^{gen} = \{s_1^{gen}, \ldots, s_N^{gen}\}$ for all the images in our dataset. We then identify the four most frequently asked questions, denoted as $\mathcal{S}^{freq} = \{s_1^{freq}, s_2^{freq}, s_3^{freq}, s_4^{freq}\}$, and use these to prompt the VLLMs. The $\mathcal{S}^{freq}$ is mainly focused on 4 aspects:

- **Number of people in the image**: Provides context for the individuals' emotional states, as those around them may influence their emotions.
- **Activity and Interactions**: Understanding individuals' actions and interactions with others can reveal more about their emotional states and the scene's context.
- **Facial Expressions**: Key indicators of emotions, providing insight into human's feelings.
- **Body Language**: Conveys mood, intentions, and complementing facial expressions.

These four types of questions are further leveraged to query the visual details, $a_i^{gen} = llava(x_i, s_i^{freq})$, where $a_i^{gen}$ is the answer provided by $llava$. The $a_i^{gen}$ is further collected into an answer set $\mathcal{A}^{gen} = \{a_1^{gen}, a_2^{gen}, a_3^{gen}, a_4^{gen}\}$.

**Scenario Self-Ask Thinking.** Going a step further, we provide the VLLM model with the image example, user question $s^{query}$, reconstructed question $s^{rec}$, and the general self-ask question-answer pairs $\mathcal{S}^{freq}$, $\mathcal{A}^{gen}$ to produce the rich scenario details description, $\mathcal{S}_i^{sce} = llava(x_i, [s^{query}, s^{rec}, \mathcal{S}^{freq}, \mathcal{A}^{gen}])$, where $[\cdot]$ denotes the concatenate operation, and $\mathcal{S}_i^{sce}$ denotes the scenario self-ask question set. Following this, the scenario self-ask answer set $\mathcal{A}^{sce} = \{a_1^{sce}, a_2^{sce}, a_3^{sce}, a_4^{sce}\}$ is generated by $a_i^{sce} = llava(x_i, [s^{query}, s^{rec}, \mathcal{S}^{freq}, \mathcal{A}^{gen}, \mathcal{S}_i^{sce}])$, where $a_i^{sce}$ is the scenario self-ask answer.

**Emotion Summarization.** After the general and scenario self-ask thinking, the critical factors impacting human emotion are comprehensively investigated, therefore, the emotion triggers can be summarized by the LLM model easily. To economically and practically summarize the emotional triggers, we leverage the recent powerful open-source LLM model, LLaMA-3, to extract them with an in-context learning scheme from all the LLaVA outputs.

**Human In-the-loop Annotation.** We use CFSA as a baseline annotation and employ LLaMA-3 for emotional trigger extraction. An example of the annotation process is depicted in Table 3. To avoid the compounding noise over CFSA five steps may introduce, four volunteers conducted a thorough manual review and detailed annotation of the entire dataset at all the phases of annotation, with three core goals: 1) Remove hallucinations generated by the VLLMs (Appendix C.1), 2) Add more commonsense knowledge to the EI process (Appendix C.2), and 3) Curate the dataset by removing unnecessary emotional triggers.

**Human Evaluation of EIBench.** We randomly selected 50 samples from each of the emotion categories, resulting in 200 samples for human evaluation, and engage 3 volunteers to assess the ground truth, rating the confidence in the emotional triggers on a scale from 0 to 5, with scores below 3 indicating errors or incompleteness in the triggers. Table 4 indicate that the quality of the emotional triggers is considered high, with all overall scores above 4.

Table 4: Human evaluation of the annotation quality on EIBench, formating are (average scores, standard deviation, [minimum, maximum]).

| Satisfaction | Happy | Angry | Sadness | Excitement | Overall |
|---|---|---|---|---|---|
| person 1 | (4.92, 0.27, [4, 5]) | (4.90, 0.30, [4, 5]) | (4.64, 0.83, [1, 5]) | (4.98, 0.13, [4, 5]) | (4.86, 0.46, [1, 5]) |
| person 2 | (4.38, 0.62, [3, 5]) | (4.62, 0.72, [2, 5]) | (3.65, 1.31, [1, 5]) | (4.58, 0.96, [1, 5]) | (4.31, 0.94, [1, 5]) |
| person 3 | (3.54, 0.63, [3, 5]) | (4.08, 0.71, [2, 5]) | (4.30, 0.75, [3, 5]) | (4.39, 0.70, [2, 5]) | (4.08, 0.70, [2, 5]) |
| average | (4.28, 0.54, [3, 5]) | (4.12, 0.98, [1, 5]) | (4.61, 0.63, [2, 5]) | (4.65, 0.69, [1, 5]) | (4.42, 0.73, [1, 5]) |

## 4.2 EVALUATION METRIC

**Emotional Trigger Recall and Long-term Coherence** Given the subjective nature of emotions, multiple triggers could elicit a particular response, and some may be missed despite thorough reviews. Therefore, we use *Recall* as one evaluation metric. If the model's interpretation overlaps with our ground truth, it is considered correct. An emotional trigger identified by the model is a true positive if it overlaps with part of our ground truth annotations; otherwise, it is a false negative. Additionally, *Long-term Coherence* in the context of EI evaluates a model's ability to maintain consistent emotional

and thematic understanding throughout extended text. This metric is crucial for tasks where the emotional narrative or flow must remain logical and coherent over multiple sentences or paragraphs. For the metric of emotional trigger recall, the LLaMA-3 or ChatGPT3.5 (gpt-3.5-turbo-0125) first extract the identified triggers from the models interpretation, then match it with the ground truth we provided. A BERT (Devlin et al., 2018) model embedding similarity between each neighboring sentence is calculated for the long-term coherence scores.

Table 5: Fine-grained emotional breakdown within primary emotional categories.

|  | Primary | Fine-grained |
|---|---|---|
| Negative | Angry | Annoyed, agitated, upset, irritated, outraged, infuriated, hostile, concerned, frustrated, serious, displeased, mad, surprised, shocked, exhibit |
|  | Sad | Forlorn, contemplative, unhappy, disheartened, dismal, solemn, sorrowful, somber, distress, miserable, discontent, upset, disappointment, distraught, displeased, frown, weary, frustration, loneliness, tragic, disappointed, melancholic, pain, injury |
| Positive | Excite | Thrill, inspired, stimulate, incite, spur, smile, happy, raised, joyful, fascinating, enjoying, brightly, spark, enthusiasm, funny, intense, pleasant, feathery |
|  | Happy | Smile, lighthearted, radiant, contented, pleased, spirited, cheerful, exhilarated, glad, blissful, energetic, joyful, optimistic, enjoying, positive, surprised |

## 4.3 DATASET OVERVIEW

Table 6: Comparison of Various Emotion Datasets. The table highlights the differences in datasets used for emotion-related tasks. ER stands for Emotion Recognition, EMER stands for Explainable Multimodal Emotion Recognition Reasoning, and EI stands for Emotion Interpretation.

| Dataset | Task | Annotator | Emotion Types | Explainable | Has Complex Label |
|---|---|---|---|---|---|
| CAER-S (Lee et al., 2019) | ER | 6 | 7 | ✗ | ✗ |
| DFEW (Jiang et al., 2020) | ER | 3 | 7 | ✗ | ✗ |
| RAF-DB (Li & Deng, 2019) | ER | 315 | 7 | ✗ | ✗ |
| HECO (Yang et al., 2022) | ER | 13 | 8 | ✗ | ✗ |
| EMOTIC (Kosti et al., 2017) | ER | - | 26 | ✗ | ✗ |
| EmoSet (Yang et al., 2023b) | ER | 10 | 8 | ✓ | ✗ |
| MER2023(EMER) (Lian et al., 2023) | EMER | 6 | 7 | ✓ | ✗ |
| EIBench | EI | 4 | 4 | ✓ | ✓ |

We chose CAER datasets with rich background information, as EI aims to interpretate emotions deeply. Facial expression recognition datasets, including close-up facial data, are unsuitable for this purpose. The CAER-S dataset (Lee et al., 2019), derived from movie clips, includes a variety of life scenarios portraying seven emotions: *angry, disgust, fear, happy, neutral, sad, and surprise*. EmoSet (Yang et al., 2023b), sourced from internet searches, is annotated with both positive emotions (*amusement, awe, contentment, excitement*) and negative emotions (*anger, disgust, fear, sadness*). Considering the inherent uncertainty of emotions and the cost for manual annotation, we selected four emotions for our initial attempt to construct an EI dataset. These include *happy* and *angry* from CAER-S, and *excitement* and *sadness* from EmoSet.

## 4.4 DATA ANALYSIS

Our benchmark extensively explores human emotions, categorizing them into four primary groups: angry, sad, excited, and happy. Each primary category is divided into fine-grained emotions. The statistics on these fine-grained emotions within each primary category are illustrated in Table 5. For instance, the anger category includes emotions such as *annoyed*, *agitated*, and *upset*, capturing varying intensities of anger. The sadness category includes emotions like *forlorn* and *contemplative*, highlighting different depths of sadness. The excitement category features emotions such as *delight* and *thrill*, reflecting different degrees of enthusiasm. The happiness category includes emotions like *lighthearted*, offering insights into various states of joy. The multifaceted complex subset consists of 50 samples, each interpreted from at least two perspectives.

**Comparison.** Table 6 compares our dataset with other emotion-related datasets. Our dataset is notable for its interpretability and complex labels. The complex subset features intricate emotions, including difficult labels absent in other datasets. Additional visualizations of our complex EI subset can be found in Appendix B.4.

Table 7: Basic EI performance of Open-Source/Close-Source Language Models, with evaluation scores presented for each subclass according to the LLaMA-3/ChatGPT criteria.

| Models | Happy | Angry | Sadness | Excitement | Overall |
|---|---|---|---|---|---|
| ***User Question*** | | | | | |
| Qwen-VL-Chat | 32.09/39.68 | 22.32/26.10 | 30.64/33.88 | 25.02/36.32 | 26.45/33.65 |
| Video-LLaVA | 55.55/53.28 | 40.42/36.97 | 50.62/45.25 | 51.78/52.23 | 49.26/47.06 |
| MiniGPT-v2 | 52.78/51.80 | **47.10/47.76** | **60.47/58.14** | 50.78/53.66 | 52.89/53.59 |
| Otter | 45.63/49.25 | 42.53/43.07 | 47.67/46.19 | 39.47/48.30 | 42.81/46.64 |
| LLaVA-1.5 (13B) | **59.01/57.52** | 45.44/41.88 | 55.16/48.64 | **57.46/58.73** | **54.37/52.20** |
| LLaVA-NEXT (7B) | 54.16/49.24 | 43.71/39.87 | 53.29/46.52 | 58.90/53.06 | 53.82/48.18 |
| LLaVA-NEXT (13B) | 57.17/55.18 | 43.16/37.93 | 54.16/45.42 | 59.38/55.29 | 54.33/48.79 |
| LLaVA-NEXT (34B) | 54.50/51.03 | 38.96/35.65 | 51.10/47.21 | 51.77/52.04 | 49.03/47.13 |
| ***User Question & Caption*** | | | | | |
| Qwen-VL-Chat | 41.94/46.34 | 32.71/31.91 | 41.82/44.16 | 38.65/43.84 | 38.47/41.54 |
| Video-LLaVA | 56.77/58.79 | 43.65/43.86 | 54.25/55.12 | 55.35/59.42 | 52.63/54.85 |
| MiniGPT-v2 | 55.11/60.04 | 47.95/51.00 | **62.29/64.24** | 51.55/57.90 | 54.05/58.37 |
| Otter | 48.97/54.67 | 34.22/37.12 | 34.57/37.55 | 35.27/42.99 | 35.62/40.85 |
| LLaVA-1.5 (13B) | 57.91/58.46 | 43.75/40.72 | 55.47/51.46 | 56.42/59.42 | 53.55/53.13 |
| LLaVA-NEXT (7B) | **64.32/61.00** | 48.60/46.74 | 58.75/53.00 | **62.99/59.39** | 58.80/54.97 |
| LLaVA-NEXT (13B) | 61.99/61.95 | **48.84/46.85** | 59.62/55.18 | 62.17/59.95 | **58.60/55.92** |
| LLaVA-NEXT (34B) | 57.51/62.73 | 46.47/47.87 | 58.35/55.84 | 60.17/59.64 | 56.60/56.24 |
| LLaMA-3 (8B) (Text Only) | 52.36/50.73 | 34.78/32.71 | 52.29/46.87 | 43.62/42.06 | 44.73/41.94 |
| ***User Question & CoT*** | | | | | |
| Qwen-VL-Chat | 41.99/44.46 | 34.62/31.06 | 43.64/39.30 | 32.78/40.04 | 36.79/38.18 |
| Video-LLaVA | 51.42/47.63 | 42.68/35.65 | 56.77/46.29 | 53.01/46.98 | 51.81/44.42 |
| MiniGPT-v2 | 56.36/57.58 | 47.71/48.32 | **59.46/56.79** | 50.21/52.39 | 52.67/53.08 |
| Otter | 49.97/51.91 | 43.23/43.71 | 50.15/46.86 | 42.30/47.16 | 45.17/46.61 |
| LLaVA-1.5 (13B) | **59.12/56.94** | 40.97/34.44 | 53.07/45.66 | 54.16/54.36 | 51.34/47.80 |
| LLaVA-NEXT (7B) | 54.74/52.04 | 44.61/41.93 | 52.69/47.63 | 52.78/47.60 | 51.14/46.66 |
| LLaVA-NEXT (13B) | 50.91/50.35 | 42.21/38.81 | 54.66/49.42 | 51.64/49.39 | 50.47/47.21 |
| LLaVA-NEXT (34B) | 52.17/49.55 | **48.35/44.45** | 55.97/50.55 | **55.29/53.46** | **53.84/50.50** |
| CFSA (LLaVA-NEXT (34B)) | 69.68/68.72 | 61.08/61.14 | 68.39/69.46 | 72.63/70.31 | 68.81/68.04 |
| ***Close-source Models*** | | | | | |
| Qwen-vl-plus[1] | 29.05/27.22 | 23.58/17.89 | 38.35/30.08 | 30.09/26.87 | 31.00/25.90 |
| ChatGPT-4V[2] | 52.30/55.74 | 48.93/48.57 | 45.00/44.42 | 46.38/49.90 | 46.86/48.58 |
| ChatGPT-4o[3] | 52.94/50.78 | 42.12/35.33 | 49.79/46.42 | 53.48/54.53 | 49.99/47.93 |
| Claude-3-haiku[4] | **59.20/60.28** | 49.87/49.84 | **67.21/63.26** | 67.55/68.10 | **63.24/62.41** |
| Claude-3-sonnet[4] | 44.58/44.45 | 38.95/42.86 | 55.98/54.40 | 61.41/62.24 | 54.10/54.89 |

**Emotion Trigger Distribution** The emotion triggers are divided into 10 categories, with definitions provided in Table 2. We also present the distribution of emotion triggers across these categories, depicted in Figure 4.4. "Atmosphere" and "Others" are the top two triggers in basic emotions, while "Social Interaction" and "Body Movements" top the list in complex emotions.

## 5 EXPERIMENTS

In this section, we evaluate the performance of both prominent open-source models and a proprietary API on our benchmark. We employed four different modes to assess the models' capabilities in EI. Following the evaluation, we present a comprehensive analysis of how each model performs in terms of EI. In the experiment, we adopted three modes to implement the EI algorithms of various models. 1) *User Question*, involves zero-shot testing of the model using the user questions, evaluating the abilities of how models deal with hu-

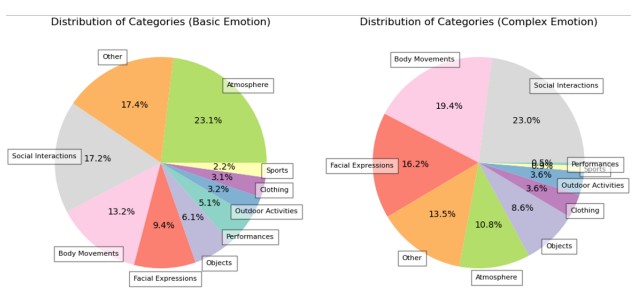

Figure 2: Visualization of the numbers of emotional triggers across different categories (Basic vs. Complex Emotions)

[1] https://qwenlm.github.io/blog/qwen-vl/

[2] https://openai.com/index/gpt-4-research/

[3] https://openai.com/index/hello-gpt-4o/

[4] https://docs.anthropic.com/en/docs/models-overview

man questions. 2) *User Question and Caption*, involves inputting questions that are parsed and restructured based on Section 4 to make the user's query more specific. Additionally, we tested the text-only model LLaMA-3 with the provided caption. 3) *CFSA setting*, uses the CFSA implemented by the LLaVA-NEXT 34B model to provide responses, showing the gap between human performance and the vllm assisted annotation dataset. And 4) *User Question and CoT setting*, inspired by the Chain of Thought (CoT) approach, involves adding the prompt "let's think step by step" after the user's question to encourage the model to reason through the problem methodically.

## 5.1 MODEL PERFORMANCE OVERVIEW

The overall performance comparison of LLMs across basic EI (Table 7) and complex EI (Table 9), reveals the distinct strengths and weaknesses of each model. In the open-source models, the LLaVA series and MiniGPT-v2 comprehend emotions well, while Qwen-VL-Chat consistently attains the lowest scores. Video-LLaVA and Otter perform moderately, although Otter shows a notable weakness in handling the excitement category. Overall, closed-source models such as ChatGPT-4 and the Claude-3 series outperform open-source models when processing user questions alone, though the Qwen-vl-plus model's performance remains subpar. Notably, the closed-source Claude-3 series (claude-3-sonnet-20240229, claude-3-haiku-20240307) excels in the basic EI setting, securing the highest overall scores and demonstrating strong EI abilities (Table 7 Close-source). However, in the complex setting, its scores are lower than those of the ChatGPT-4 series (Table 9). Table 10 reports the long-term coherence scores between models, showing that their scores are close, demonstrating a consistent ability to maintain context over comprehending emotion.

Inspried by PsychoBench (Huang et al., 2023), we assigned the LLM with different personas to test if they have different performance in EIBench. We implement the best performance LLM in Table 7 for 4 personas settings: 1) without persona assigned, 2) an helpful AI assistant, 3) expert in other domain (architecture), and 4) expert in emotion understanding. Results in Table 8 show that assigning the LLM with the expert in emotion understanding persona improve its ability in EI task, while as an expert in architecture may cause a little decrease than without persona or default as an ai assistant.

Table 8: Models performance under different persona prompting. The score are LLaMA-3/ChatGPT evaluation.

| Model | w/o Persona | AI Assistant | Architecture | Emotion |
|---|---|---|---|---|
| LLaVA-NEXT (7B) | 52.09/46.64 | 49.48/46.13 | 45.32/38.40 | **53.82/48.18** |
| LLaVA-NEXT (13B) | 52.44/50.07 | 49.69/48.12 | 44.26/35.79 | **54.33/48.79** |
| LLaVA-1.5 (13B) | 51.58/53.62 | 51.04/50.66 | 49.58/43.16 | **54.37/52.20** |
| Claude-3-haiku | 58.28/58.62 | 60.37/59.86 | 31.81/25.53 | **63.24/62.41** |

The CFSA scores reveal that while EI is challenging for models, they can successfully identify 68% of emotional triggers. By converting the problem into a series of simple VQA tasks, the scores show a significant improvement. Additionally, the scores indicate that the VLLM assisted annotation, still lag behind human-level annotations, highlighting the considerable effort by our manually labeling.

## 5.2 ABILITIES COMPARISON

In the direct User Question setting, all models scored relatively low. After adding the Caption, the scores of all models improved, but it is notable that the Otter model's overall score decreased by approximately 7%. MiniGPT-v2 scored higher in the Angry and Sadness categories, while the LLaVA series models performed better overall, with LLaVAv1.5 (13B) achieving the highest scores, particularly excelling in the Happy and Excitement categories. Interestingly, increasing the model size did not lead to better performance; the 34B model's scores even declined. The Qwen-VL-Chat model performed poorly across all emotion categories, and its performance with Caption and user questions was even worse than that of the text-only LLaMA-3 model.

We introduced "let's think step by step" as part of the input, leveraging the CoT approach to improve model performance. Results show that this method consistently outperformed the direct User Question setting, indicating the complexity of the EI task. Detailed reasoning and step-by-step responses proved more effective than direct answers, helping uncover more emotional triggers. This finding aligns with our observations using the CFSA method, where models better identified emotion triggers through detailed, step-by-step analysis.

Table 9: Evaluation of the complex EI ability among the VLLMs.

| Models | Recall |
|---|---|
| ***Open-Souce*** | |
| Qwen-VL-Chat | 22.00/32.40 |
| Video-LLaVA | 30.90/32.27 |
| MiniGPT-v2 | 35.10/36.00 |
| Otter | 27.90/33.23 |
| LLaVA-1.5 (13B) | 38.10/39.53 |
| LLaVA-NEXT (7B) | 38.71/33.50 |
| LLaVA-NEXT (13B) | 39.16/33.60 |
| LLaVA-NEXT (34B) | 35.37/33.10 |
| ***Close-Source*** | |
| Qwen-vl-plus | 20.37/19.60 |
| Claude-3-haiku | 24.00/24.77 |
| Claude-3-sonnet | 21.37/22.40 |
| ChatGPT-4V | 28.00/30.60 |
| ChatGPT-4o | **39.27/39.57** |

Table 10: Metric of Long-term Coherence between VLLMs in user question setting.

| Models | Coherence |
|---|---|
| ***Open-Souce*** | |
| Qwen-VL-Chat | 84.49 |
| Video-LLaVA | 84.89 |
| MiniGPT-v2 | 84.70 |
| Otter | 85.03 |
| LLaVA-1.5 (13B) | 84.50 |
| LLaVA-NEXT (7B) | 81.02 |
| LLaVA-NEXT (13B) | 81.09 |
| LLaVA-NEXT (34B) | 84.96 |
| ***Close-Source*** | |
| Qwen-vl-plus | 83.00 |
| Claude-3-haiku | **85.98** |
| Claude-3-sonnet | 84.53 |
| ChatGPT-4V | 81.97 |
| ChatGPT-4o | 80.65 |

Table 9 shows the performance of models on the multifaceted complex subset. The performance of open-source models is similar to that on the basic subset, but their scores are significantly lower. This subset evaluates the models' abilities in multifaceted emotional reasoning and empathy. Even the highest-scoring model, LLaVA-1.5, only achieves 38.10/39.53 points, while is notably close to the best-performing closed-source model, ChatGPT-4. Notably, the Claude-3 series, which performed best on the basic subset, does not achieve SOTA results on this subset and even scores lower than some open-source models. This indicates that while the Claude-3 series excels at basic EI, its ability to handle more complex, multifaceted emotional reasoning is less effective.

## 6 CONCLUSION

In this paper, we introduce the Emotion Interpretation (EI) task, which focuses on interpreting emotional triggers rather than merely labeling emotions. We establish the Emotion Interpretation Benchmark (EIBench) using a VLLM-assisted construction method, Coarse-to-Fine Self-Ask (CFSA), consisting of 1,615 basic EI samples, and 50 well-annotated multifaceted complex EI samples. Extensive experiments evaluating commonly used open-source and closed-source models demonstrate that these models have limited proficiency in the EI task. Some are even performing lower than the text-only LLaMA-3 model in the caption-provided setting. This task not only enhances EI but also provides a metric for evaluating the emotional intelligence of VLLMs. By considering emotions from multiple perspectives, it aids in analyzing implicit emotions and advances the field of emotion recognition. Additionally, our annotation method facilitates emotion reasoning dataset development and provides a valuable resource for multi-turn dialogue in emotion research.

## 7 LIMITATION AND SOCIAL IMPACT

Interpreting emotions is inherently challenging due to the subjective nature of personal experiences. It is an open-world problem that requires a nuanced understanding, which is difficult to achieve with current methodologies. Therefore, we encourage the research community to explore innovative approaches to enhance our collective comprehension of emotional diversity. Our annotation process, which is augmented by VLLM-assisted tools, is conducted by a team of four annotators. While this approach provides valuable insights, it may also introduce potential biases. The significant costs associated with large-scale annotation in EIBench have constrained the range of emotions we can explore as well. Besides, EIBench at this time is only for the purpose of assessment, not fine-tuning. To mitigate these issues, we would expand our pool of annotators to include a more diverse group of individuals from various backgrounds and underrepresented minority groups, as well as increase the scale of EIBench for support training. This expansion will not only enrich our dataset but also ensure that our understanding of emotions is more inclusive and representative of the broader human experience.

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

# A BASELINE MODELS

## A.1 OPEN-SOURCE MODELS

### A.1.1 QWEN-VL-CHAT

Qwen-VL-Chat Bai et al. (2023) is a multimodal large language model (LLM)-based AI assistant developed by Alibaba Cloud. It supports flexible interactions, including multiple image inputs, multi-round question answering, and the use of bounding boxes for grounding. It employs a 448x448 resolution visual encoder, which enhances fine-grained text recognition, document question answering, and bounding box annotation.

Qwen-VL-Chat supports English, Chinese, and multilingual conversations, enabling end-to-end recognition of bilingual text in images. Furthermore, Qwen-VL-Chat can manage multi-image interleaved conversations, facilitating the input and comparison of multiple images. This functionality also allows for the specification of questions related to the images and the capability to engage in multi-image storytelling.

### A.1.2 VIDEO-LLAVA

Video-LLaVA Lin et al. (2023) serves as a baseline for the LVLM, adept at handling both images and videos simultaneously. It begins by aligning the representations of images and videos into a unified visual feature space. By doing so, Video-LLaVA allows for the mutual enhancement of image and video processing within a single visual representation framework, outperforming models specifically designed for either images or videos alone.

### A.1.3 MINIGPT-V2

MiniGPT-v2 Chen et al. (2023a) is a sophisticated multimodal language model that stands out for its unified interface capability for diverse vision-language tasks such as image description, visual question answering, and visual grounding. Its architecture simplifies the integration of high-resolution visual inputs with a large language model, utilizing a technique that concatenates four neighboring visual tokens, significantly reducing the sequence length and enhancing training efficiency.

The model's excellence is attributed to its unique three-stage training strategy. Initially, it undergoes pretraining with a broad mix of datasets to establish a robust foundation in vision-language understanding. This is followed by a multi-task training phase, where the model refines its capabilities on specific tasks using fine-grained datasets, excluding weakly-supervised data to focus on high-quality image-text alignment. The final stage involves multi-modal instruction tuning and chatbot enhancement, integrating complex datasets to improve the model's conversational skills and its ability to handle diverse instructions, thus preparing it for real-world applications.

### A.1.4 OTTER

Otter Li et al. (2023b) is an advanced model crafted to facilitate multi-modal in-context instruction tuning, leveraging the OpenFlamingo Awadalla et al. (2023) framework. This framework adeptly conditions the language model on associated media, such as an image complementing a caption or an instruction paired with a response. The training of Otter is grounded in the Multi-Modal In-Context Instruction Tuning (MIMIC-IT) Li et al. (2023a) dataset, which presents each data instance as an instruction-image-answer triplet enriched with pertinent in-context examples. Through this approach, Otter acquires the proficiency to adeptly follow instructions, drawing insights from the contextual learning exemplars provided.

### A.1.5 LLAVA-1.5

LLaVA-1.5 Liu et al. (2024b) is an improved baseline model that uses CLIP-ViT-L-336px Radford et al. (2021) with an MLP projection and adds academic-task-oriented VQA data with response formatting prompts to the LLaVA model. Compared to LLaVA, LLaVA 1.5 improves its model performance by using an MLP cross-modal connector and incorporating academic task-related data such as VQA. LLaVA-1.5 13B checkpoint uses merely 1.2M publicly available data.

### A.1.6 LLaVA-NEXT

Compared to LLaVA-1.5, LLaVA-NEXT Liu et al. (2024a) has improved reasoning, OCR, and world knowledge. It was designed to have a high resolution with an aim to preserve its data efficiency. The model's capacity to perceive intricate details in an image is significantly improved and reduces model hallucination that conjectures the imagined visual content. LLaVA-NEXT is trained on High-quality User Instruct Data and Multimodal Document or Chart Data. LLaVA-NEXT also considers more LLM backbones such as Mistral-7B Jiang et al. (2023) and Nous-Hermes-2-Yi-34B[1].

## A.2 CLOSE-SOURCE MODELS

### A.2.1 QWEN-VL-PLUS

Qwen-vl-plus is Qwen's Enhanced Large Visual Language Model. Significantly upgraded for detailed recognition capabilities and text recognition abilities, supporting ultra-high pixel resolutions up to millions of pixels and arbitrary aspect ratios for image input. It delivers significant performance across a broad range of visual tasks. And it only supports online API calling.

### A.2.2 CLAUDE-3

The Claude-3 model, developed by Anthropic, emphasizes safety, controllability, and ethical considerations, setting it apart from OpenAI's ChatGPT. Claude employs adversarial training to enhance robustness and mitigate harmful outputs, focusing heavily on reducing biases and ensuring fairness. It also aims for greater transparency and interpretability, providing detailed documentation to help users understand the decision-making process.

In contrast, while ChatGPT also addresses safety and ethical concerns, its design may not be as extensively focused on these aspects as Claude. ChatGPT excels in general-purpose NLP tasks like text generation, translation, and summarization, making it highly versatile. However, Claude's rigorous emphasis on security and ethical standards makes it particularly suitable for applications requiring high safety and ethical compliance.

### A.2.3 CHATGPT-4

ChatGPT-4 (ChatGPT-4o, ChatGPT-4V), developed by OpenAI, is a state-of-the-art language model known for its versatility and performance across a wide range of natural language processing tasks. Leveraging extensive pre-training on diverse datasets, ChatGPT-4 excels in text generation, conversation, translation, summarization, and more. It builds upon the strengths of its predecessors, incorporating advanced techniques to enhance coherence, relevance, and fluency in its outputs.

OpenAI has made significant efforts to improve the safety and ethical considerations of ChatGPT-4. The model includes mechanisms to reduce harmful and biased outputs and employs user feedback to continually refine its performance. Additionally, OpenAI provides extensive documentation and guidelines to help users understand and effectively use the model. While ChatGPT-4 is highly adaptable and powerful in general NLP applications, it also strives to meet high standards of safety and ethical compliance, making it a robust tool for a variety of use cases.

## B EIBENCH

### B.1 DATASET APPLICATION

The EIBench dataset, with its unique focus on EI, offers a wide array of applications across various fields. Here, we outline several key areas where EIBench can be particularly impactful:

1. **Emotion Recognition Systems**: EIBench can significantly enhance the development of emotion recognition systems by providing a nuanced understanding of emotional triggers. Unlike traditional datasets that merely label emotions, EIBench allows models to learn the underlying causes of emotions, thereby improving the accuracy and depth of emotion

---

[1]https://huggingface.co/NousResearch/Nous-Hermes-2-Yi-34B

recognition systems. This can be particularly useful in applications such as customer service bots, mental health diagnostics, and interactive entertainment, where understanding the root cause of emotions can lead to more *empathetic* responses.

2. **Human-Computer Interaction (HCI)**: In HCI, understanding user emotions is crucial for creating responsive and adaptive interfaces. EIBench can be employed to train systems that better comprehend user emotions and adjust their interactions accordingly. For instance, in virtual assistants or interactive gaming, recognizing why a user feels a certain way can lead to more personalized and satisfying user experiences.

3. **Psychological and Behavioral Research**: The dataset provides a rich resource for researchers studying the dynamics of emotional responses. By analyzing the emotional triggers annotated in EIBench, researchers can gain insights into common emotional patterns and the factors that influence them. This can contribute to better therapeutic approaches in clinical psychology and enhance our understanding of human behavior.

4. **Social Media Analysis**: EIBench can enhance sentiment analysis tools used in social media monitoring by providing a deeper understanding of the emotional context behind posts. This can be useful for brands and organizations to gauge public sentiment more accurately, respond appropriately to customer feedback, and manage their online presence more effectively.

## B.2 TARGET AUDIENCES

This benchmark is designed to advance the field of EI by fostering a more flexible and nuanced understanding of emotions. Recognizing the subjective nature of human emotions, we have established this multifaceted and complex EI benchmark. Successfully addressing the challenges presented by EIBench can lead to the development of empathetic AI systems, thereby enhancing human-computer interaction and emotion-sensitive applications. Furthermore, we anticipate that a multifaceted approach to emotion will benefit implicit emotion recognition tasks, including humor understanding and harmful stance detection.

## B.3 BASIC EIBENCH

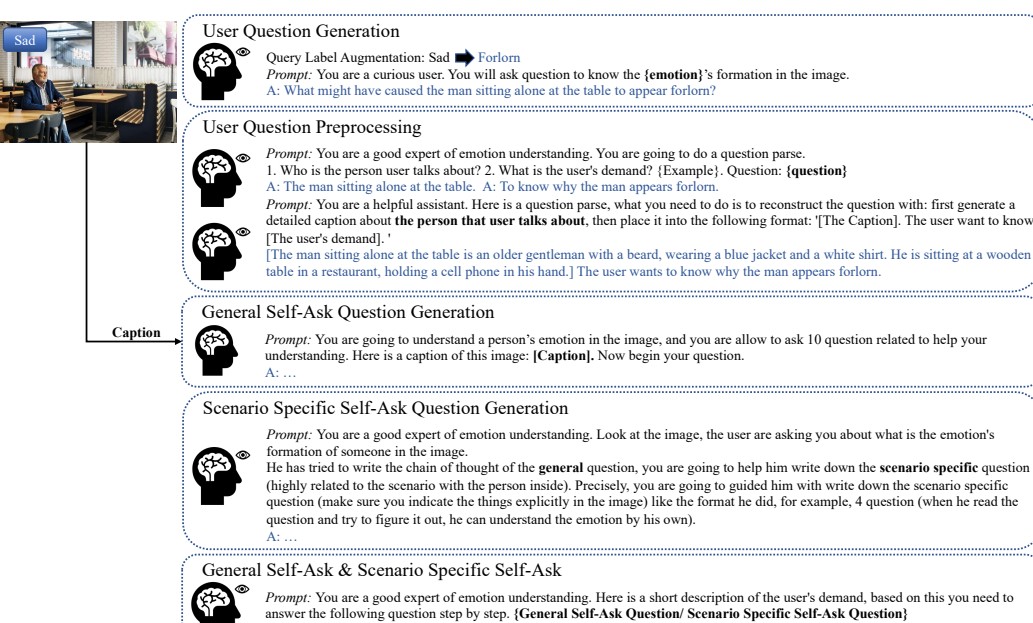

Figure 3: Pipeline of the VLLM-assisted dataset construction.

Table 11: Visualization of basic EI dataset, an image is corresponded to one user questions.

**Examples of the Basic EI Dataset**

| User Question | *What led to the formation of the arouse to the man in this image?* |
| Emotional Trigger | 1. Climbing a steep, snow-covered slope. 2. Physical effort and concentration. 3. Potential hazards and challenges. 4. Cold environment. 5. Determination to reach the goal. |

| User Question | *What do you think might have caused the person's delight as they look out the window?* |
| Emotional Trigger | 1. Snowy scene outside the car. 2. Smile on her face. 3. Enjoying the view. 4. Serenity of the winter environment. 5. Excitement of experiencing a snowy day. 6. Personal or emotional connections to snowy weather or winter scenes. 7. Fresh snowfall, brightness of the snow reflecting sunlight, or peacefulness of the scene. |

| User Question | *What do you think might have caused the man holding the box in the image to become lighthearted?* |
| Emotional Trigger | 1. Holding the "Uberweiss" box. 2. Smiling. 3. Friendly and approachable body language. 4. Positive and relaxed atmosphere of the laundry room. 5. Interaction with others in the laundry room. |

| User Question | *What might have caused the woman in the image to appear content and happy?* |
| Emotional Trigger | 1. Positive news about her health. 2. Pleasant interaction with a medical professional. 3. Comforting conversation with a friend or family member. 4. Good news about her health. 5. Positive relationship with the medical staff. |

| User Question | *What might have caused the woman in the image to appear irritated or angry?* |
| Emotional Trigger | 1. Service issue (mistake in order, long wait, problem with payment process). 2. Unpleasant environment (noise levels, cleanliness, presence of other customers). 3. Dissatisfaction with food or service. 4. Frustration or annoyance with the conversation or situation. |

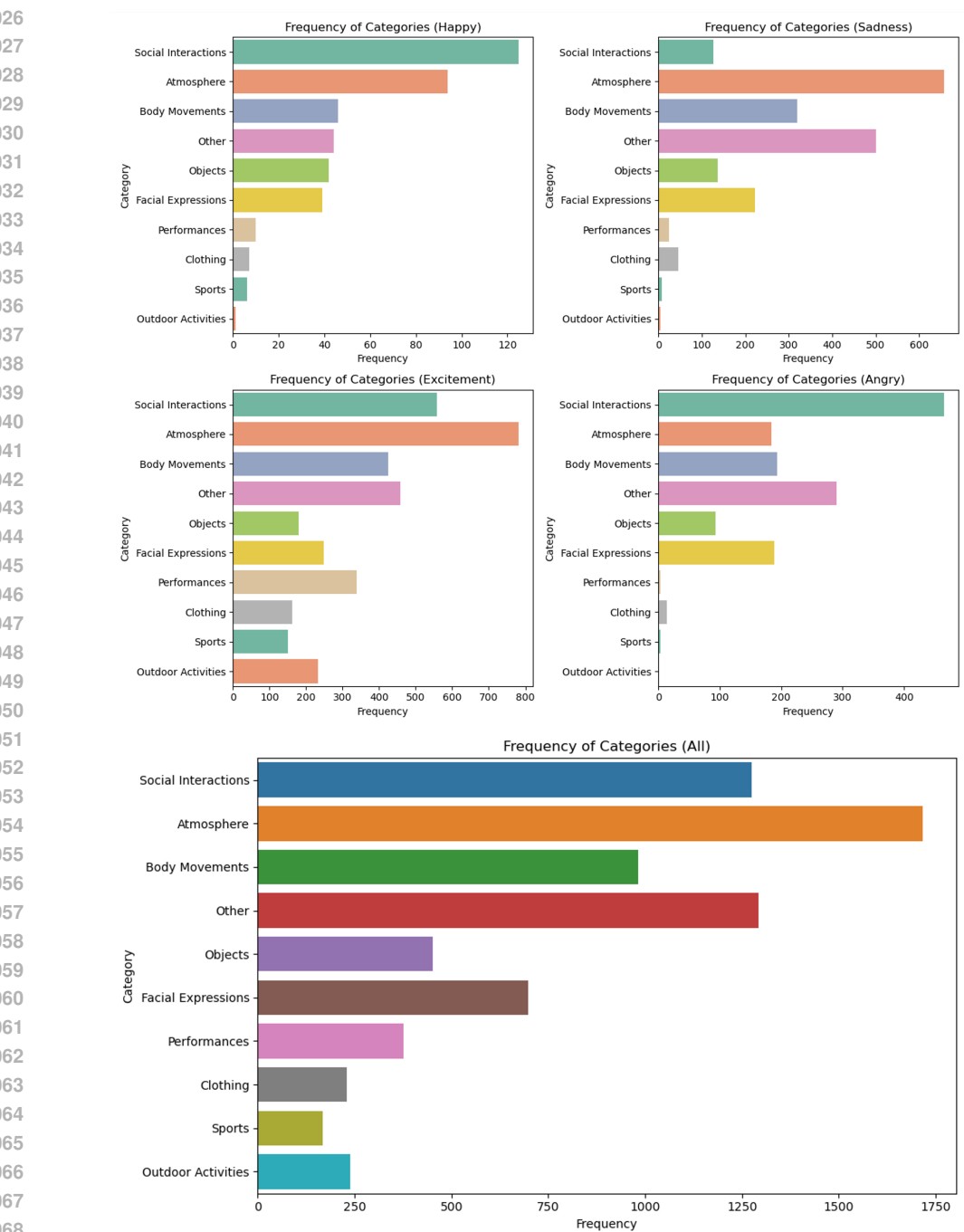

Figure 4: Visualization of the numbers of emotional triggers across different categories (Basic Emotions).

Table 12: Statistics of the Emotional Trigger Types (Basic Emotions).

| Atmosphere | Social Interactions | Body Movements | Facial Expressions | Objects | Performances | Outdoor Activities | Clothing | Sports | Other |
|---|---|---|---|---|---|---|---|---|---|
| 23.11% | 17.17% | 13.24% | 9.40% | 6.07% | 5.06% | 3.20% | 3.08% | 2.25% | 17.41% |

## B.4 COMPLEX EI SUBSET

Table 13: Visualization of complex EI subset, an image is corresponded to multiple user questions.

**Examples of the Complex EI Subset**

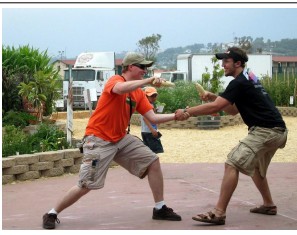

| | |
|---|---|
| User Question (1) | *Why does the kid in the background seem excited?* |
| Emotional Trigger | 1. Head turning back. 2. Starring at the two playing with each other on the focus. 3. Sense of motion from the event. 4. Maybe excited about the desire to join them. |
| User Question (2) | *What do you think might have caused the kid in the background of the image to be confused?* |
| Emotional Trigger | 1. Head turning back. 2. Two others acting abnormally. 3. Two others each holding a stick of corn. 4. Maybe curious about the event. 5. Maybe wondering about the motivation for the abnormality. |

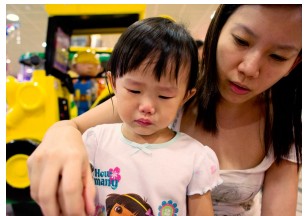

| | |
|---|---|
| User Question (1) | *What may caused the little girl upset?* |
| Emotional Trigger | 1. Crying. 2. Can not making handiwork. 3. The woman blamed her. |
| User Question (2) | *What may caused the little girl happy?* |
| Emotional Trigger | 1. Crying but the women comfort her. 2. Can not making handiwork. 3. Woman help her finishing the work. |
| User Question (3) | *What may cause the woman angry?* |
| Emotional Trigger | 1. The girl is not obedient. 2. The girl can't do handiwork. 3. The girl can't learn no matter how much taught. 4. Step-by-step instruction. |

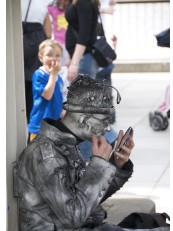

| | |
|---|---|
| User Question (1) | *Why does the baby show the fear expression?* |
| Emotional Trigger | 1. The man's scary outfit. 2. Afraid of the man. 3. The man's makeup. 4. Covering mouth with hand. |
| User Question (2) | *What make the baby surprise and happy?* |
| Emotional Trigger | 1. Shocking face and gesture. 2. Staring at someone. 3. Sense of unbelievable. 4. A man colored in silver on the focus. 5. Maybe shocked to see something abnormal. |

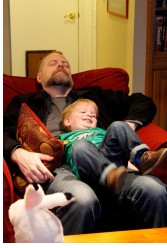

| | |
|---|---|
| User Question (1) | *Why does this man in the picture look exhausted and annoyed?* |
| Emotional Trigger | 1. Maybe lack of Sleep. 2. Closed-eyes. 3. Taking care of a young child. 4. Tired of the child. 5. Naughty child. |
| User Question (2) | *Why does this man being enjoyment and pleasure?* |
| Emotional Trigger | 1. Enjoying spending time with his child. 2. Child lying in arms. 3. Satisfied with the moment. 4. Sense of company of family. 5. Engaging in playful activities. |

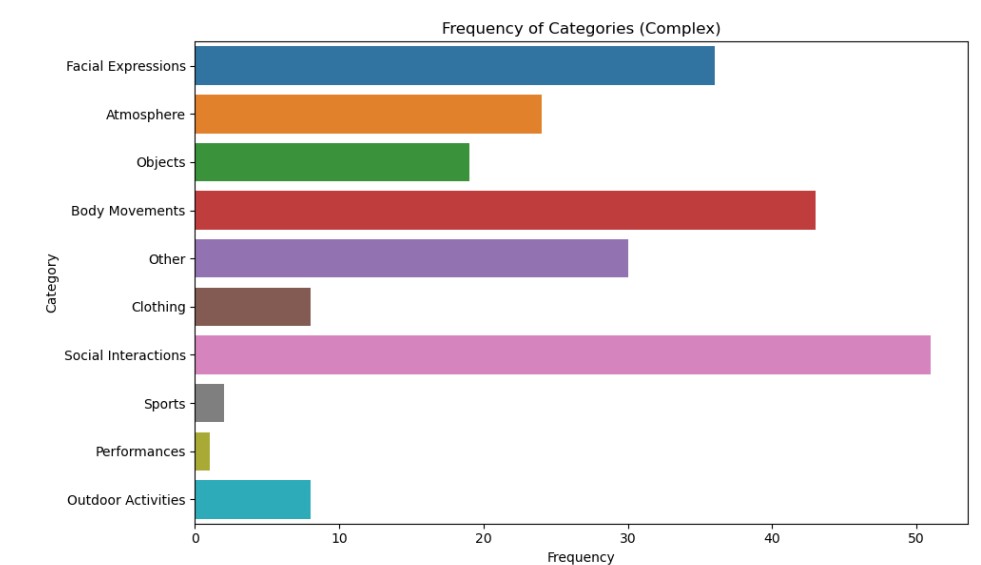

Figure 5: Visualization of the numbers of emotional triggers in the Complex EI Subset.

Table 14: Statistics of the Emotional Trigger Types (Complex Emotions).

| Atmosphere | Social Interactions | Body Movements | Facial Expressions | Objects | Performances | Outdoor Activities | Clothing | Sports | Other |
|---|---|---|---|---|---|---|---|---|---|
| 10.81% | 23.00% | 19.37% | 16.22% | 8.55% | 0.45% | 3.60% | 3.60% | 0.9% | 13.51% |

## C  HUMAN IN THE LOOP DATA CLEANING DETAILS

### C.1  HALLUCINATIONS IN VLLMS

In this section, we present examples of hallucinations in Vision Large Language Models (VLLMs), along with the human-in-the-loop data cleaning process to address them. Table 15 provides examples of hallucinated emotional triggers generated by VLLMs in response to user questions. The examples illustrate instances where VLLMs generate triggers that do not accurately reflect the visual context of the images, or are not present in the image. For instance, in the first example, the VLLM hallucinates "Doing mountain biking" as the trigger for the man's participation in the outdoor activity, despite no evidence of biking in the image. Removing these hallucinations reduces bias in our datasets introduced by VLLMs.

Table 15: Example of Hallucinations in VLLMs. Hallucinations are indicated in red, while other text is indicated in gray.

**Examples of the Human Cleaning Process of Hallucinations**

| User Question | *What might have motivated the man in the image to participate in this outdoor activity, given his gear and the environment?* |
|---|---|
| Emotional Trigger (Raw) | 1. Determination and concentration. 2. Challenge of the race or trail. 3. Personal goals. 4. Desire to improve mountain biking skills. 5. Well-prepared gear. 6. Environmental factors (rocky slope, weather conditions). 7. Doing mountain biking. |

| User Question | *What could have caused the man in the image to appear outraged or hostile?* |
|---|---|
| Emotional Trigger (Raw) | 1. Holding a black bag. 2. Animated conversation or gesture. 3. Furrowed eyebrows. 4. Open mouth. 5. Wide or squinting eyes. 6. Leaning forward or gesturing with hands. 7. Brown couch (as a place where he typically relaxes or discusses matters) |

| User Question | *What might have caused the man in the image to be angry or upset?* |
|---|---|
| Emotional Trigger (Raw) | 1. KANO CAP ABILITY sign on the wall. 2. Feeling overwhelmed or pressured by his workload. 3. Undervalued or overworked in his professional role 4. Recent events or interactions in the workplace that have caused stress or frustration. 5. Tension and stress in his body language (posture, grip on the mug). |

| User Question | *What might have caused the man in the image to appear angry or frustrated?* |
|---|---|
| Emotional Trigger (Raw) | 1. Disagreement with a family member. 2. Concern about a meal he is preparing. 3. Problem at work that he is thinking about while in the kitchen. 4. Serious or intense mood due to work-related issue or concern. |

## C.2 EXAMPLES OF DATA CLEANING FOR COMMONSENSE KNOWLEDGE

In this section, we provide examples of the human-in-the-loop data cleaning process to instill commonsense knowledge into the dataset. Table 16 presents instances where commonsense knowledge is incorporated to improve the emotional trigger identification. By incorporating commonsense knowledge, we enhance the accuracy and realism of emotional triggers, improving the overall quality of the dataset for EI tasks.

Table 16: The Human in the Loop process instills Commonsense Knowledge into the dataset. Text orange represents added commonsense knowledge.

**Examples of Data Cleaning for Commonsense Knowledge**

| User Question | *What might have caused the baby's delight in this image?* |
|---|---|
| Emotional Trigger | 1. Halloween costume and bib with a pumpkin design. 2. Interaction with the person holding them up. 3. Festive atmosphere and attention from the person holding them up. 4. First Halloween experience. |

| User Question | *What led to the excitement on the woman's face?* |
|---|---|
| Emotional Trigger | 1. A toy written "Beijing Welcome". 2. Taking a photo with Tienanmen Square. 3. First time to Beijing. |

| User Question | *What might have caused the man in the image to become excited and make a funny face?* |
|---|---|
| Emotional Trigger | 1. Celebratory event or milestone related to the year 2021. 2. Excitement and joy. 3. Playful or lighthearted moment shared between the man and the woman. 4. Achievement or personal milestone. 5. Festive and celebratory atmosphere. |

| User Question | *Why does the kid in the background seem excited?* |
|---|---|
| Emotional Trigger | 1. Head turning back. 2. Starring at the two playing with each other on the focus. 3. Sense of motion from the event. 4. Maybe excited about the desire to join them. |

# D  CASE STUDY OF THE VLLMS EI ABILITIES

Table 17: Case Study of the VLLMs EI Abilities, each example present the EI process and the LLaMA-3/ChatGPT evaluation.

| Examples of VLLMs EI | |
|---|---|
| |  |
| User Question | *What factors might be contributing to the anger of the man in the image?* |
| Emotional Trigger | 1. Staring with wide opening eyes. 2. Fully aggressive vibe. 3. Metal claws. 4. Defending gesture. 5. Shrunk muscle. |
| Qwen-VL-Chat | The man in the image is holding several metal claws in his hands and making a fierce face, which suggests that he is angry. Factors that might be contributing to his anger could include a conflict or disagreement with someone, a frustrating situation, or a sense of injustice or betrayal. It's also possible that the man is simply expressing his emotions in a playful or dramatic way for fun or to entertain others... |
| LLaMA-3 Eval | {score: 2/5}. *Correctly identified: The man's facial expression and the metal claws in his hands are correctly identified as potential triggers for his anger (points 2 and 3). *Missed: The other factors mentioned in the ground truth label, such as staring with wide opening eyes, defending gesture, and shrunk muscle, are not mentioned in the record. |
| ChatGPT Eval | {score: 3/5}. Explanation: The record correctly identified the metal claws and the defending gesture as potential triggers of anger. However, it missed identifying the staring with wide opening eyes, fully aggressive vibe, and shrunk muscle as additional potential triggers. |
| Video-LLaVA | The man in the image appears to be angry, possibly due to the presence of multiple knives in his hands. These knives could be causing him distress or discomfort, or he might be experiencing frustration or aggression. Additionally, the man's facial expression and body language suggest that he is feeling overwhelmed or threatened. It is important to note that the man's emotions may be influenced by multiple factors, such as personal circumstances, the environment, or the presence of others around him. |
| LLaMA-3 Eval | {score: 1/5}. The record correctly identifies one potential trigger for the man's emotion, which is the presence of multiple knives in his hands. However, the other triggers mentioned, such as facial expression, body language, and personal circumstances, are not explicitly mentioned in the ground truth label. The ground truth label specifically mentions staring with wide opening eyes, metal claws, defending gesture, and shrunk muscle, which are not mentioned in the record. |
| ChatGPT Eval | {score: 2/5}. Explanation: The record correctly identified the aggressive vibe and defending gesture as potential triggers for the emotion of anger. However, it missed identifying the staring with wide opening eyes, metal claws, and shrunk muscle. |
| MiniGPT-v2 | The factors contributing to the anger of the man in the image could be his physical posture, facial expression, and hand gesture. His raised hands with their fingers interlocked, forming a claw-like shape, suggest that he is expressing aggression or hostility. His furrowed brow and intense gaze further reinforce this interpretation. Additionally, his clenched fists, which are positioned in front of his chest, indicate that he might be experiencing strong emotions such as frustration, irritation, or even rage. These elements together suggest that the man in the image may be expressing a range of negative emotions, which can contribute to his overall appearance of anger. |
| LLaMA-3 Eval | {score: 2/5}. The record correctly identified the following potential triggers: *Raised hands with fingers interlocked, forming a claw-like shape (matches ground truth label 3: Metal claws). *Furrowed brow and intense gaze (matches the overall aggressive vibe described in ground truth label 2: Fully aggressive vibe). The record missed the following potential triggers: *Staring with wide opening eyes (not mentioned in the record). *Defending gesture (not explicitly mentioned, but the record does mention hand gesture, which could be interpreted as a defensive gesture, but it's not clear). *Shrunk muscle (not mentioned at all in the record). *Clenched fists (while mentioned, it's not explicitly linked to the emotions described in the ground truth label, which is frustration, irritation, or rage). The record provides a good analysis of the physical posture, facial expression, and hand gesture, but misses some specific details mentioned in the ground truth label. |
| ChatGPT Eval | {score: 3/5}. The record correctly identified the factors of physical posture, facial expression, and hand gesture contributing to the man's anger. The raised hands forming a claw-like shape, furrowed brow, and intense gaze were correctly interpreted as signs of aggression or hostility. However, the record missed identifying the wide opening eyes and fully aggressive vibe as triggers of anger. The interpretation of defending gesture and shrunk muscle were not mentioned in the record. |
| Otter | Several factors might contribute to the man's angry expression in this image. He could be upset due to the intense lighting shining on his face, which might make him appear angrier than he actually is. Additionally, the man is holding a pair of large scissors, which could be potentially intimidating or making him feel uncomfortable. The close-up nature of the image might also emphasize his facial expressions, accentuating his anger. Furthermore, the context of the scene could be a photo shoot or a situation where the man might be experiencing emotions related to the situation he is in. |
| LLaMA-3 Eval | {score: 2/5}. The record correctly identifies two potential triggers for the man's angry expression: the intense lighting shining on his face and the close-up nature of the image emphasizing his facial expressions. However, it misses three potential triggers mentioned in the ground truth label: the man's staring with wide opening eyes, the fully aggressive vibe, and the metal claws. |
| ChatGPT Eval | {score: 1/5}. Explanation: The record correctly identified the potential trigger related to the man holding a pair of large scissors, which could be potentially intimidating or making him feel uncomfortable. However, it missed the other triggers mentioned in the ground truth label, such as staring with wide opening eyes, fully aggressive vibe, metal claws, defending gesture, and shrunk muscle. |
| ChatGPT-4V | The man in the image is showing a facial expression and body language that can be interpreted as anger or aggression. Key features contributing to this interpretation may include: 1. Eyebrows: His eyebrows are furrowed, creating wrinkles on his forehead, which is a common indicator of anger or concentration. 2. Eyes: His eyes have an intense gaze which suggests focus and potential hostility. 3. Mouth: His mouth is closed tightly, which can be a sign of tension or suppressed emotion. 4. Facial muscles: There is a general tension in his facial muscles, adding to the stern expression. 5. Stance: His body stance appears confrontational, with his arms and hands in front of him as if ready for combat. 6. Props: The metallic claws he's holding, reminiscent of the character Wolverine from the X-Men series, suggest a readiness for defense or attack, fitting with the theme of aggression or anger... |
| LLaMA-3 Eval | {score: 4/6}. The record correctly identified the following potential triggers: *Eyebrows: Furrowed eyebrows are a common indicator of anger or concentration, which is correct. *Eyes: An intense gaze can suggest focus and potential hostility, which is correct. *Mouth: A closed mouth can be a sign of tension or suppressed emotion, which is correct. *Facial muscles: General tension in facial muscles adds to the stern expression, which is correct. *Props: The metallic claws suggest a readiness for defense or attack, fitting with the theme of aggression or anger, which is correct. The record missed the following potential triggers: *Stance: The ground truth label mentions a "defending gesture", which is not explicitly mentioned in the record. *Muscle: The ground truth label mentions "shrunk muscle", which is not mentioned in the record... |
| ChatGPT Eval | {score: 3/5}. Explanation: The record correctly identified the intense gaze in the eyes, the confrontational stance, and the presence of metal claws as potential triggers of aggression or anger. However, it missed mentioning the wide opening eyes and the defending gesture as described in the ground truth label. The mention of shrunk muscles was not identified in the record. |
| **Continue on next page** | |

| | Examples of VLLMs EI |
|---|---|
| **Emotional Trigger** | 1. Staring with wide opening eyes. 2. Fully aggressive vibe. 3. Metal claws. 4. Defending gesture. 5. Shrunk muscle. |
| **LLaVA-1.5 (13B)** | The man in the image, who is dressed in a flannel shirt and jeans and wearing a belt, appears to be angry. Several factors might be contributing to his anger. One possible factor is that he is posing with a set of fake Wolverine claws, which could indicate that he is a fan of the X-Men character Wolverine and is expressing his frustration or disappointment about the character's portrayal in movies or other media. Another factor could be that he is simply in a bad mood or experiencing personal issues that are unrelated to the claws. It is also possible that the man is posing for a humorous or satirical photo, which might be intended to provoke a reaction or make people laugh... |
| **LLaMA-3 Eval** | {score: 2/5}. The record correctly identifies two potential triggers for the man's anger: the fake Wolverine claws (factor 3) and the possibility that he is a fan of the X-Men character Wolverine and is expressing frustration or disappointment (factor 1). However, it misses three other potential triggers mentioned in the ground truth label: the man's staring with wide opening eyes (factor 1), the fully aggressive vibe (factor 2), and the defending gesture (factor 4). |
| **ChatGPT Eval** | {score: 3/5}. Explanation: The record correctly identified the metal claws and the defending gesture as potential triggers for the emotion of anger. However, it missed identifying the staring with wide opening eyes, fully aggressive vibe, and shrunk muscle as additional potential triggers. |
| **LLaVA-NEXT (7B)** | In the image, you see a person with a stern expression, holding a set of six stainless steel claws that resemble those of Wolverine from the "X-Men" comics and movies. The factors contributing to the anger of the man in the image could include: 1. **Stress or Discomfort**: The claws might be causing him physical discomfort or pain, which can lead to an angry expression. 2. **Frustration**: He might be frustrated with a situation or a person, and the claws could be a symbol of his anger. 3. **Emotional Exhaustion**: The act of holding the claws might be emotionally exhausting, and the man could be angry because he feels overwhelmed. 4. **Performance or Role**: The man could be an actor or a cosplayer portraying a character with anger, and the claws are part of his costume. 5. **Personal Issues**: There could be personal issues or conflicts that are causing him anger, and the claws could be a metaphor for his feelings of aggression or frustration. 6. **Artistic Expression**: The image might be staged for an artistic purpose, and the man's expression and the claws are part of a creative concept. 7. **Reaction to a Situation**: The man might be reacting to a recent event or situation that has caused him anger, and the claws are a representation of his emotional response. It's important to note that without additional context, it's difficult to pinpoint the exact cause of his anger. However, these are some plausible factors that could be contributing to his expression. |
| **LLaMA-3 Eval** | {score: 1/7}. My assessment is that 2 out of the 7 potential triggers identified in the record match the ground truth label. The two correct triggers are: * **Frustration**: The ground truth label mentions a fully aggressive vibe, which aligns with the idea that the man might be frustrated with a situation or a person. * **Reaction to a Situation**: The ground truth label mentions a defending gesture, which suggests that the man might be reacting to a recent event or situation that has caused him anger. The other 5 potential triggers do not directly match the ground truth label. |
| **ChatGPT Eval** | {score: 1/5}. Explanation: The record correctly identified the presence of metal claws, which is one potential trigger for the emotion. However, it missed the other triggers such as staring with wide opening eyes, fully aggressive vibe, defending gesture, and shrunk muscle as indicated in the ground truth label. |
| **LLaVA-NEXT (13B)** | The image shows a man with a stern and intense expression, holding up what appear to be claw-like objects. His facial expression, body language, and the context of the image suggest several factors that might be contributing to his anger: 1. **Personal Distress**: The man's facial expression, with furrowed brows and a tight-lipped mouth, indicates that he is experiencing significant emotional distress. This could be due to personal issues, frustration with a situation, or a perceived injustice. 2. **Physical Discomfort**: The claw-like objects he is holding could imply that he is experiencing physical pain or discomfort, which might be contributing to his anger. 3. **Perceived Threat**: The claw-like objects might be interpreted as a weapon or a defensive posture, suggesting that the man feels threatened or under attack, which could be a source of his anger. 4. **Role or Character**: If the man is portraying a character from a movie or a role-playing scenario, his anger might be part of the character's persona or the narrative of the scene. 5. **Context of the Photo**: The context of the photo, such as the setting or the people around him, could be contributing to his anger. For example, he might be in a situation where he feels cornered or unsupported. 6. **Emotional Trigger**: There could be an event or situation that has triggered his anger, which is not visible in the image... |
| **LLaMA-3 Eval** | {score: 3/6}. My assessment is based on the record's identification of potential triggers for the man's anger. The record correctly identifies the following triggers: 1. **Personal Distress**: The record mentions the man's facial expression, which is consistent with the ground truth label of "staring with wide opening eyes", indicating a sense of distress or alarm. 2. **Physical Discomfort**: The record mentions the claw-like objects, which is consistent with the ground truth label of "metal claws", suggesting physical discomfort or pain. 3. **Perceived Threat**: The record mentions the claw-like objects as a potential source of the man's anger, which is consistent with the ground truth label of "fully aggressive vibe" and "defending gesture", indicating a perceived threat. The record misses the following triggers: 1. **Role or Character**: The record does not explicitly mention the man's role or character, which is mentioned in the ground truth label. 2. **Context of the Photo**: The record does not provide specific context about the photo, which is mentioned in the ground truth label. 3. **Emotional Trigger**: The record does not provide a specific event or situation that triggered the man's anger, which is mentioned in the ground truth label. 4. **Shrunk muscle**: The record does not mention the man's physical posture or muscle tension, which is mentioned in the ground truth label... |
| **ChatGPT Eval** | {score: 2/5}. Explanation: The record correctly identified the potential triggers of "Physical Discomfort" and "Perceived Threat" based on the man's expression and the objects he is holding. However, it missed identifying triggers related to "Staring with wide opening eyes," "Fully aggressive vibe," and "Defending gesture" as described in the ground truth label. |
| **ChatGPT-4o** | Based on the image, several factors might be contributing to the man's expression of anger or intensity: 1. **Body Language:** The man is clutching blades that are extending from his hands, which might symbolize aggression or readiness for battle, contributing to a perception of anger or intensity. 2. **Facial Expression:** His furrowed brows, intense eye contact, and tight lips create an expression commonly associated with anger or determination. 3. **Costume and Pose:** The attire and the claws suggest he might be cosplaying a character known for aggression or conflict, which might be contributing to his intense demeanor. These elements together create an impression of anger or a serious mood, likely for dramatic effect, possibly related to the character he is portraying. |
| **LLaMA-3 Eval** | {score: 3/5}. My assessment is based on the record's identification of potential triggers contributing to the man's expression of anger or intensity. Here's a breakdown of the correct and missed triggers: Correctly identified triggers: 1. **Body Language:** The record correctly notes the man's clutching of blades as a symbol of aggression or readiness for battle, which contributes to a perception of anger or intensity. 2. **Facial Expression:** The record accurately identifies the man's furrowed brows, intense eye contact, and tight lips as an expression commonly associated with anger or determination. 3. **Costume and Pose:** The record correctly notes the attire and claws suggesting the man might be cosplaying a character known for aggression or conflict, which contributes to his intense demeanor. Missed triggers: 1. **Staring with wide opening eyes:** The record does not mention the man's eye expression, which is a key aspect of his intense demeanor. 2. **Fully aggressive vibe:** The record does not explicitly state the man's overall aggressive vibe, although it does mention the costume and pose suggesting aggression. 3. **Defending gesture:** The record does not mention the man's defending gesture, which is an important aspect of his intense demeanor. 4. **Shrunk muscle:** The record does not mention the man's muscle tone, which is not explicitly stated in the ground truth label. Overall, the record correctly identifies three out of five potential triggers contributing to the man's expression of anger or intensity. |
| **ChatGPT Eval** | {score: 3/5}. Explanation: The record correctly identified the factors related to body language, facial expression, and costume that contribute to the perception of anger or intensity. However, it missed identifying the specific details of staring with wide opening eyes and defending gesture as mentioned in the ground truth label. |
| **Continue on next page** | |

| Examples of VLLMs EI | |
|---|---|
| Emotional Trigger | 1. Staring with wide opening eyes. 2. Fully aggressive vibe. 3. Metal claws. 4. Defending gesture. 5. Shrunk muscle. |
| LLaVA-NEXT (34B) | The image you've provided shows a person with claws protruding from their hands, which is a characteristic feature of the fictional character Wolverine from the X-Men series. The person is also making a stern facial expression, which could be interpreted as a sign of anger or aggression. Factors that might contribute to the anger or intense expression in the image could include: 1. Role Playing: The person might be a fan of the character Wolverine and could be role-playing or imitating the character's typical intense demeanor. 2. Costume or Prop: The claws could be part of a costume or prop for a cosplay event or a themed party, where the person is trying to embody the character's persona. 3. Performance: The person could be an actor or performer engaging in a theatrical or filmic portrayal of the character, and the expression is part of the character's emotional state in that scene. 4. Personal Expression: The person might be expressing their own emotions, perhaps frustration, anger, or determination, and is using the claws and intense expression as a metaphor or a way to convey their feelings. 5. Artistic Choice: The image could be a piece of art or a still from a video where the artist or performer has chosen to portray the character in a specific emotional state for creative or storytelling purposes. It's important to note that without additional context, it's difficult to determine the exact reason for the person's expression. The factors listed above are speculative and based on the visual cues present in the image. |
| LLaMA-3 Eval | {score: 2/5}. My assessment is based on the record provided, which attempts to understand the emotion in the image by identifying potential triggers. The record correctly identifies two factors that might contribute to the anger or intense expression in the image: 1. Role Playing: The record correctly notes that the person might be a fan of the character Wolverine and could be role-playing or imitating the character's typical intense demeanor, which aligns with the ground truth label's description of a "fully aggressive vibe". 2. Performance: The record also correctly suggests that the person could be an actor or performer engaging in a theatrical or filmic portrayal of the character, which is consistent with the ground truth label's description of a "defending gesture". However, the record misses three potential triggers: 1. Staring with wide opening eyes: The record does not mention this specific visual cue, which is present in the ground truth label. 2. Metal claws: While the record correctly notes the presence of metal claws, it does not explicitly link them to the emotion or expression. 3. Shrunk muscle: The record does not mention this specific detail, which is present in the ground truth label. Overall, the record provides some insightful observations, but could benefit from more specific and detailed analysis to accurately capture the emotion and expression in the image. |
| ChatGPT Eval | {score: 1/5}. Explanation: The record correctly identified the presence of metal claws in the image, which aligns with the ground truth label. However, it missed identifying the other emotional triggers such as staring with wide opening eyes, fully aggressive vibe, defending gesture, and shrunk muscle. The analysis focused more on speculative factors related to role-playing, costume, performance, personal expression, and artistic choice rather than accurately pinpointing the specific emotional cues present in the image. |
| Qwen-vl-plus | Based on the image alone, it is difficult to determine with certainty why the man appears angry or aggressive. However, there could be several reasons for his expression: - He may have had a frustrating experience that triggered negative emotions. - The pose and stance he has adopted suggest confidence and assertiveness which can also come across as aggression. It's important to note that interpreting human emotions from images can often lead to misinterpretations due to cultural differences, personal biases, and lack of context. Therefore, any conclusions drawn should not be taken too seriously without additional information about the situation. |
| LLaMA-3 Eval | {score: 0/5}. The record does not identify any potential triggers correctly. The record provides a general analysis of the image, mentioning possible reasons for the man's expression, but does not specifically identify any triggers that match the ground truth label. The ground truth label includes specific details such as staring with wide opening eyes, metal claws, and shrunk muscle, which are not mentioned in the record. |
| ChatGPT Eval | {score: 0/5}. Explanation: The record of understanding did not correctly identify any of the potential triggers mentioned in the ground truth label. The reasons provided for the man's expression were more general and did not align with the specific details mentioned in the ground truth. |
| LLaMA-3 (8B) | Based on the image alone, it is difficult to determine with certainty why the man appears angry or aggressive. However, there could be several reasons for his expression: - He may have had a frustrating experience that triggered negative emotions. - The pose and stance he has adopted suggest confidence and assertiveness which can also come across as aggression. It's important to note that interpreting human emotions from images can often lead to misinterpretations due to cultural differences, personal biases, and lack of context. Therefore, any conclusions drawn should not be taken too seriously without additional information about the situation. |
| LLaMA-3 Eval | {score: 0/5}. The record does not identify any potential triggers correctly. The record provides a general analysis of the image, mentioning possible reasons for the man's expression, but does not specifically identify any triggers that match the ground truth label. The ground truth label includes specific details such as staring with wide opening eyes, metal claws, and shrunk muscle, which are not mentioned in the record. |
| ChatGPT Eval | {score: 0/5}. Explanation: The record of understanding did not correctly identify any of the potential triggers mentioned in the ground truth label. The reasons provided for the man's expression were more general and did not align with the specific details mentioned in the ground truth. |

