# OpenReview forum: "EIBench: Assessing the Emotion Interpretation ability of Vision Large Language Models"
_ICLR.cc/2025/Conference — ICLR 2025 Conference Withdrawn Submission_

### Official Review · Reviewer_zq7q · 2024-11-03

**Soundness:** 2
**Presentation:** 2
**Contribution:** 2
**Rating:** 5
**Confidence:** 4

**Summary:**

This paper introduces the Emotion Interpretation (EI) task and EIBench, advancing emotion recognition research in AI. EI task shifts focus from emotion classification to interpreting emotional triggers, enabling deeper understanding of human emotional responses.

Authors also develop a VLLM-assisted data annotation technique that combines efficient machine annotation with human verification.

The work aims to foster more empathetic AI systems by providing a structured framework for understanding emotions in context, contributing significantly to human-computer interaction advancement.

**Strengths:**

- The EI task proposes a novel approach to emotion recognition, focusing on understanding emotional triggers.
- The paper attempts to evaluate various open-source and closed-source models under different settings.

**Weaknesses:**

This paper presents several notable limitations that warrant discussion:

1. **Limited Practical Impact Due to Absence of Fine-tuning Paradigm**
  - The work focuses solely on evaluating EI capabilities of existing models without providing methods to enhance their performance
  - The lack of fine-tuning guidelines or frameworks significantly limits the practical value for the research community

2. **Inconsistent Emotion Classification Taxonomy**
  - The fine-grained emotion categorization in Table 5 contains logical inconsistencies. For example, 'happy' is inappropriately classified under the 'excite' primary category rather than 'Happy'
  - Such classification errors suggest insufficient validation of the emotional taxonomy structure

3. **Questionable Value of Fine-grained Classification**
  - The practical contribution of the fine-grained emotion classification system is unclear. Experimental results are predominantly reported using the four coarse-grained categories. The granular categorization appears superfluous given its limited use in the actual analysis. The work would benefit from either:
    * Demonstrating concrete applications of the fine-grained classification
    * Focusing more deliberately on the core four-category system

These limitations suggest opportunities for future work to enhance the practical impact and theoretical foundation of emotion interpretation tasks.

**Questions:**

Please see weaknesses.

---

### Official Review · Reviewer_Pf69 · 2024-11-04

**Soundness:** 2
**Presentation:** 3
**Contribution:** 2
**Rating:** 5
**Confidence:** 4

**Summary:**

1. The paper introduces the Emotion Interpretation task, which involves predicting the triggers that led to the emotion in the image.
2. The authors develop EIBench, a benchmark for this task, using their proposed method Coarse-to-Fine Self-Ask (CFSA).
3. The CFSA method involves breaking user questions into a series of simple VQA queries, that are fed to the VLLM.
4. The paper evaluates the recall and coherence of various open-source and closed-source models on this benchmark under various settings.

**Strengths:**

1. The paper introduces a novel task, Emotion Interpretation (EI), where the task is to understand the triggers behind emotions.
2. The authors ensure reliability by conducting an extensive manual cleaning process to prevent hallucinations and other inaccuracies.
3. The paper conducts a thorough evaluation of both open-source and closed-source models across multiple settings, including zero-shot, caption-provided, reasoning with Chain of Thought (CoT), and various personas.

**Weaknesses:**

1. The dataset is relatively small, with only 1,615 images, of which just 50 are complex images. It focuses on four primary emotions (happy, angry, sad, and excited), which may not capture the full range of human emotional experiences. Expanding the dataset to include a broader range of emotions, such as fear, disgust, surprise, and neutral states, would be beneficial.
2. The paper lacks a comprehensive human study to evaluate the identified triggers. Given the subjective nature of emotions, evaluating with only three volunteers seems insufficient, especially if the volunteers share the same demographics. Additionally, it would be helpful to provide the guidelines given to the volunteers.
3. The metrics used in the paper are not very clear and may not accurately capture the correctness of the model's responses.

**Questions:**

1. What is the purpose of categorizing emotional triggers? Are these categories used in any step of the CFSA? How are the triggers classified into these categories—through human input or an LLM?
2. For the Human-in-the-Loop annotation, what qualifies as an "unnecessary emotional trigger"?
3. I'm not sure I fully understand the suitability of the evaluation metric used. Recall is calculated by comparing model outputs with ground truth annotations from Llama-3/ChatGPT, and if there’s a match with the ground truth, it’s considered correct. Since these triggers are subjective, how do you ensure the metric's suitability, especially given that CFSA might miss certain triggers?
4. Could you provide a comparison of model performance with human performance on the EI task? This would help clarify the gap between human and AI capabilities and make the figures in the tables more meaningful.
5. What specific measures were taken to ensure the quality and consistency of the human annotations? Were any inter-annotator agreement metrics applied to validate these annotations?
6. In Table 4, how are the min and max for the average row computed? Why is the minimum of the average row 1 while the min for other rows are 4,2,2 in the Angry column.
7. Regarding the long term coherence scores, "model embedding similarity between each neighboring sentence" is calculated between which sentences? Are these sentences the triggers themselves?

---

### Official Review · Reviewer_V8B1 · 2024-11-04

**Soundness:** 2
**Presentation:** 1
**Contribution:** 2
**Rating:** 5
**Confidence:** 4

**Summary:**

1. This paper introduces the EI (Emotion Interpretation) task, i.e. predicting emotional triggers for an image with given emotions, which is evaluated using Recall and Coherence.
2. The authors introduce the EIBench benchmark, for EI consisting of images with 1615 samples for basic and 50 complex emotions.
3. The paper introduces a VLLM assisted annotation scheme Coarse to Fine Self Ask (CFSA) for preliminary annotation.
4. The authors do an extensive evaluation of open and close-source models on the EI task under different settings.

**Strengths:**

1. EI is a novel task to interpret cues or triggers for the evoked emotion in the image.
2. The authors conduct human annotation for the entire dataset on the outputs of CFSA, and the quality of the EIBench
3. The evaluation for EI is extensive including open and close source models
4. The EI bench consists of images from fine-grained emotions, and multi-faceted emotions as well.

**Weaknesses:**

1. **Dataset Size and Composition**:
   - 1,615 samples with only 50 classified as complex may not provide sufficient coverage for the number of categories, especially as prior literature [1] considers multiple important factors like colors and objects in emotion elicitation. To ensure robustness, Emotion-Image (EI) samples should cover a more comprehensive combination of these factors.
   - The current set of emotion categories omits primary emotions such as Love and Fear, which are essential in widely accepted emotion taxonomies. Adding these would provide a more complete emotional spectrum.

2. **Fine-Grained Emotion Overlap**:
   - While the inclusion of fine-grained emotions is appreciated, some labels are inconsistently assigned across categories, resulting in overlap. For example:
     - "Upset," "Frustration," "Displeased," "Enjoying," and "Joyful" appear in multiple categories.
     - Emotions from opposing categories, such as "Surprised," are present in both "Negative Angry" and "Happy."
   - **Clarification**: It would be helpful to know if these overlaps were intentional and whether samples were included in both classes during dataset construction.

3. **Human Evaluation**:
   - The evaluation of dataset quality relies on only three annotators, which may not be sufficient. To strengthen the validity:
     - Consider including inter-annotator agreement scores to assess consistency.
     - Provide more details on the annotators' backgrounds and qualifications, as this could impact the dataset quality and the interpretability of the annotations.

4. **Dataset Collection Details**: Important aspects of dataset construction and filtering are missing, including:
     - How were images selected from CAER and EmoSet?
     - What process was used to obtain the user questions?
     - The use of the feature "X_face" suggests only images with faces were considered—please confirm or clarify.
     - The paper mentions "critical factors impacting human emotion are comprehensively investigated." It would be helpful to specify what these critical factors are.
     - Specify which version of LLaMA-3 was used for summarization in your experiments.
     - In Table 4, was an average applied to minimum and maximum values, or were they rounded? Global minimum and maximum values would provide clarity, and these specifics should be included in the caption.

5. **Writing and Clarity**:
   - **Table Captions**: Captions need to clarify the contents better. For example, Table 2 lacks context—is it analysis, input, or output?
   - **Metric Definition**: "We use Recall as one evaluation metric. If the model’s interpretation overlaps with our ground truth." Please clarify what is meant by "overlap" in this context, as this may not be an ideal metric for a generative task.
   - **Model Specifications**: For the Recall metric, specify the LLaMA version (e.g., Chat/Base, 7B, 70B) used to maintain clarity.

**Questions:**

1. Where are the details for IRB/ERB approval for the annotations?
For remaining questions please refer to weaknesses.

---

### Official Review · Reviewer_qqDf · 2024-11-05

**Soundness:** 2
**Presentation:** 3
**Contribution:** 2
**Rating:** 3
**Confidence:** 3

**Summary:**

This paper proposes a new task named Emotion Interpretation to interprets the reasons behind emotions, and creates the EIBench Benchmark using an effective VLLM-assisted data annotation technique. The EIBench includes 1,615 basic and 50 multi-faceted complex emotion interpretation samples. They also evaluate several LLMs on the EIBench, highlighting the limitations of current models in interpreting emotions.

**Strengths:**

1. The paper highlights an interesting problem in emotion interpretation, specifically focusing on identifying emotional triggers rather than just classifying emotions.
2. Despite its small size, the introduction of the EIBench dataset represents an effort to address the lack of labeled data specifically designed for fine-grained emotion interpretation.
3. The paper is well-organized and has a clear structure. The clarity and logical flow of the presentation make the paper easy to understand.

**Weaknesses:**

1. The primary contribution of this paper is proposing a new task on emotion interpretation, claiming it differs from existing mainstream tasks (such as emotion cause extraction) by focusing on interpreting emotional triggers rather than merely classifying emotions. However, this is not a novel task. Emotion-Cause Analysis in [1] and Multimodal Emotion Cause Generation in [2] both consider emotion categories as well as fine-grained cause interpretation (rather than traditional cause extraction), which is almost identical to the formalized definition in the paper. It is recommended that the authors thoroughly review related literature and compare their proposed task with works like Emotion-Cause Analysis and Multimodal Emotion Cause Generation.

- [1] EMO-KNOW: A Large Scale Dataset on Emotion-Cause (EMNLP 2023)
- [2] Observe before Generate: Emotion-Cause Aware Video Caption for Multimodal Emotion Cause Generation in Conversations (MM 2024)

2. The secondary contribution of the paper is introducing a labeled dataset for emotion cause interpretation. However, EIBench has a very small data size (fewer than 2,000 samples). Additionally, the authors have not made the dataset publicly available and have not analyzed or quantified key indicators such as data quality and diversity. Therefore, compared to existing datasets (such as the large and open-source datasets EMO-KNOW [1] and ECGF [2]), the contribution of this dataset is minimal.

3. The experimental conclusions are insufficient. The authors conducted preliminary experiments on some LLMs only in a zero-shot setting without evaluating them in few-shot, supervised fine-tuning, or reinforcement learning (weak supervision) settings. Therefore, the conclusion "Experiments show limited proficiency of existing models in EI" is unreliable. It is recommended that the authors expand the experimental results in few-shot, supervised fine-tuning, and reinforcement learning (weak supervision) settings to verify the accuracy of their conclusions.

4. The authors propose an effective VLLM-assisted data annotation technique for the EIBench dataset, but they have not compared it with existing annotation methods to demonstrate its advantages.

5. The experimental setup in the paper lacks detail. For example, there is no explanation of the rationale behind the selection of model types and sizes used.

**Questions:**

1. Compared to existing Emotion-Cause Analysis in [1] and Multimodal Emotion Cause Generation in [2], what are the differences and advantages of the task proposed in this paper?

2. Will the proposed dataset be open-sourced? Why wasn't the sample size of the dataset increased? Has there been any data analysis conducted on the collected dataset, such as evaluating data quality and diversity? Why did the authors choose the current coarse-grained emotion labeling system instead of a finer-grained annotation system like GoEmotions?

3. In the experimental setup, why didn’t the paper use settings such as few-shot, supervised fine-tuning, or reinforcement learning (weak supervision)? Considering the randomness and instability in outputs from large models, did the paper conduct statistical significance testing for the reported results?

---

### Official Review · Reviewer_HPcX · 2024-11-05

**Soundness:** 1
**Presentation:** 1
**Contribution:** 2
**Rating:** 3
**Confidence:** 4

**Summary:**

The paper makes the following contributions:
1. Defining a new task, Emotion Interpretation
2. Coarse-to-Fine Self-Ask (CFSA) method using VLLMs
3. developing the EIBench dataset, which includes 1,615 basic EI samples, and 50 multifacets complex EI samples
4. Evaluation of several models on the new task


I have the following questions and suggestions for this work:

1. Please illustrate the differences in the following tasks, perhaps using a running example: EI, ECE, HS, and EMER.

1. Overall, this is a dataset paper but I found it hard to understand the paper in several places, and critical details are missing, which are important for method verification, validation, and reproduction. Particularly, details in the following sections are either missing or incomplete:

    - L271 - Why did you choose only 4 questions? Please provide an analysis for this.
    - L287-289 - where do the user questions come from? Which sets of users?
    - L301-304 and L306-310 - what is the annotation protocol? What is the inter-annotator agreement?
    -  L319-321 - why not precision as a target metric?
    - Tables 4 and 5 - how did the authors choose 4 emotions? Which schema is it based on? How did the authors combine secondary emotions with the primary ones?
    -  L355-360 - it is hard to understand how CAER-S and EmoSet are used for EI and how this section relates to the previous sections, particularly 4.1.1.

1. In several places, the grammar seems incorrect, which breaks the flow of reading. (L137-140, L158-161, L172-175, etc). Please try to correct them.

1. The proposed dataset is too small (only 1600 samples).



Overall, primarily due to a lack of clarity around the presentation of the work, I would recommend another round of reviews for this work.

**Strengths:**

Indicated in the summary section above

**Weaknesses:**

Indicated in the summary section above

**Questions:**

Indicated in the summary section above

---

### Official Review · Reviewer_Qgvm · 2024-11-05

**Soundness:** 2
**Presentation:** 1
**Contribution:** 3
**Rating:** 5
**Confidence:** 3

**Summary:**

The paper introduces a new benchmark for emotion cause understanding entitled Emotion Interpretation Benchmark (EIBench). EIBench utilizes an annotator and model-in-the-loop annotation process to enrich two existing emotion recognition datasets with meaningful interpretations (i.e., causes) of emotions by querying various LLMs using a series of visual question answering prompts. First, the input is preprocessed and a caption to the input image is generated, followed by prompting an LLM with various pre-defined questions aimed at identifying general information from the input. Next, emotion-rich information is extracted using the previously extracted information and the emotion connotation of the input, followed by using a summarization approach to identify the triggers from the outputs of the prior steps. To ensure qualitative responses, at each step, annotators verify and correct the generated answers. The paper carries our comprehensive experiments to measure the performance of various baselines in multiple setups on the introduced benchmark.

**Strengths:**

Reasons to accept

- EIBench can be leveraged to measure the ability of LLMs to interpret emotions. Given the complexity of the task, I see it very useful for hillclimbing by identifying loss patterns and improving the capability of language models.
- The proposed annotator and model-in-the-loop approach is interesting, significantly reducing the labeling costs that would be needed for labeling for causes from scratch.
- The experimental setup is solid. Numerous baselines are considered with different prompting techniques and in various settings (e.g., with/without caption, with/without CoT).
- The paper carries out interesting analyses by studying the different types of triggers or performance analyses when prompting with different personas.

**Weaknesses:**

Reasons to reject

- This is a dataset paper but there is little focus on the annotation process and interpreting the data. It is critical to extend the discussion of the data considerably. This is even more critical since a significant portion of the emotion causes/interpretations are model-generated. Here are important insights that are missing: (1) Are the final responses (i.e., triggers) natural in terms of language? (2) How many changes did the annotators need to make in each stage? How many hallucinations were there? How many times did an extra trigger need to be added? Is there a specific emotion whose triggers were harder to identify? (3) What is the qualification of the annotators? What was the process to train these annotators? How do we know we can trust these annotations? How many annotators were used per example? (4) In L306-L316 there is a human evaluation of EIBench. Why is there such a big variance in scores between the annotators. Are these the same annotators as L299-L305? (5) Why only 4 general questions were used? Have you tried fewer/more? How does this affect the quality of the data? Overall, I think the paper should focus more on the quality of data and the processes used to ensure this quality. Additionally, each step of the annotation process needs a thorough analysis: lexical/language analysis.
- The clarity and quality of the writing could be significantly improved. Grammatical errors and unsuitable phrasings make the paper difficult to read.
- Emotion-cause extraction (ECE) has less real-world applicability compared to emotion-cause pair extraction [1] (ECPE). The evaluation lacks ECPE results.

Minor:
-  Size of the data (1615 examples) is somewhat limited.


[1] Xia and Ding, Emotion-Cause Pair Extraction: A New Task to Emotion Analysis in Texts


Overall, I think EIBench can be extremely useful. I am worried about the quality, the coverage of emotion triggers, as well as the "naturalness" of the interpretations. I am willing to raise my scores based on questions 1, 2, 3, 4.

**Questions:**

See questions 1, 2, 3, 4.

---

### Note · Authors · 2024-11-26

I have read and agree with the venue's withdrawal policy on behalf of myself and my co-authors.